# Selective observation following betrayal shapes the social inference landscape

**Sangkyu Son** [1,2]*, **Seng Bum Michael Yoo** [1,2,3,4]*

**1** Center for Neuroscience Imaging Research, Institute for Basic Science (IBS), Suwon, Republic of Korea, **2** Department of Biomedical Engineering, Sungkyunkwan University, Suwon, Republic of Korea, **3** Department of Intelligent Precision Healthcare Convergence, Sungkyunkwan University, Suwon, Republic of Korea, **4** Department of Neurosurgery, Baylor College of Medicine, Houston, Texas, United States of America

* ss.sangkyu.son@gmail.com (SS); sbyoo@g.skku.edu (SBMY)

## Abstract

Despite limited access to others' actions and outcomes, humans excel at inferring hidden intentions. Given only partial access, how do they decide what to observe, and how does selective observation shape inference? Here, we examined how choosing what to observe can bias the inference about others' intentions. Participants played a game where they pursued a fleeing target while a computerized opponent acted competitively or cooperatively. Participants overestimated the opponent's competitiveness after the opponent acted more competitively than expected, whereas no such bias occurred when the opponent was more cooperative than expected. This asymmetry depended on the sequence of events, resembling hysteresis, a form of path dependence observed in physical systems. We found that these biases became stronger when participants chose to observe the opponent instead of their own avatar, and this choice came at the cost of losing precise control over their avatar. Our findings highlight the trade-off in selecting what to observe, as the resulting inference biases propagate differently depending on the interaction history.

## Author summary

We often think that everything necessary for understanding others is already visible. However, in reality, we see only a small part of what others do. Focusing on one specific cue while ignoring others—like a gesture versus facial expression—can lead to misunderstandings about their intentions. How do we decide what to observe and how does that decision shape our understanding of others' intentions? These questions have been a long-standing challenge in social inference and, more broadly, theory of mind. To address this, we question the common assumption that all social cues are freely accessible. Instead, we propose that observing others is limited by available resources and, thus, comes with a cost—focusing on one cue means paying less attention to others. This idea of partial

**Data availability statement:** The code generated and/or analyzed during the current study is available at https://github.com/SangkyuSon/socialObservationHysteresis. The datasets generated and/or analyzed during the current study are available at https://doi.org/10.5281/zenodo.17986013.

**Funding:** This research was supported by Ministry of Food and Drug Safety of South Korea, RS-2024-00333012 (SS and SBMY). The funders had no role in study design, data collection and analysis, decision to publish, or preparation of the manuscript.

**Competing interests:** The authors have declared that no competing interests exist.

observability helps explain how a specific sequence of negative past experiences can shape lasting biases in inferring others' intentions. Our study builds on previous models by relaxing the assumption that we see everything others do, and offers a more realistic view that people understand others based on what they choose to observe.

## Introduction

In a social context, one must infer the latent attributes of agents, such as others' intentions or goals, from a series of observations. For example, when we see an adult following a child, we can infer the adult's intention, either to prevent the child from falling as a parent or to bypass them as a stranger, by observing a set of actions over time. This inverse process for inferring latent social attributes or others' mental states through observing others' actions and following outcomes is called *social inference* [1–3]. Despite its complexity, humans can seamlessly infer latent social attributes (e.g., roles in the scene) even from animated shapes, as demonstrated in seminal developmental psychology studies [4,5].

In contrast to predicting actions from one's known goal (i.e., forward problem), social inference is often conceptualized as an inverse problem of deducing one's latent values or goals from a series of observed actions [2,3]. From a Bayesian perspective, this can be formalized as inferring the posterior probability of others' goals given the observed actions (p(goal|action)) [3]. Similarly, this can be viewed as inverse reinforcement learning, where social inference involves learning others' state or action value functions through action sequences [2,6]. Both emphasize the chain-like inference over time as a Markov Decision Process (MDP), where action sequences serve as the key clues for inferring others' unobservable goals or values.

While most focus has been on action per se, less attention has been given to the role of observation in sampling those actions. This neglect is attributable to the implicit assumption that one observation could capture all available information, similar to a fully observable Markov decision process (MDP). However, observation is inherently partial and selective—agents do not perceive every action equally, but selectively allocate their gaze to sample socially prioritized aspects of action at each moment. Even among observed information, the socially pertinent attributes can be re-prioritized along the visual pathway [7–9], including simple shape-based actions [10]. This suggests that social inference models should expand upon the previous inverse MDP framework and encompass how limited access to action can shape social inference.

Therefore, we re-conceptualize social inference as a bidirectional interaction between the self and others, involving the dynamic unfolding of actions and observations (Fig 1). The key idea is that much of the world beyond one's own actions and observations is at least partially observable or fully hidden. So, one must strategically allocate the limited observational resources across the environment, others' actions, and what they observe. For example, in observational learning, the learner attends

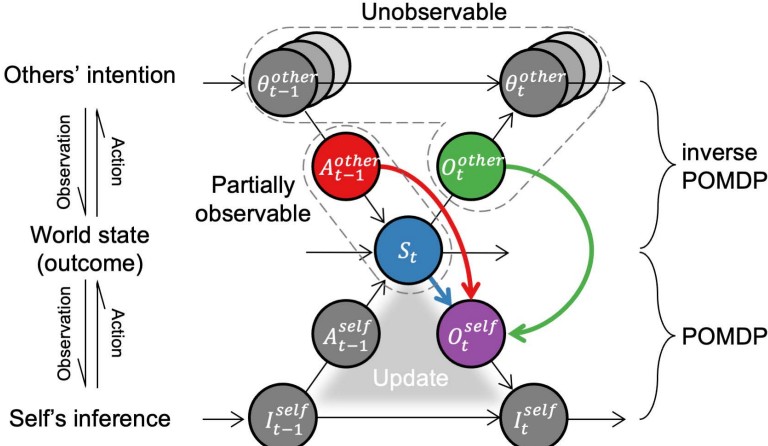

**Fig 1. A graphical framework for understanding social inference. Social inference is a bidirectional process between self and others in the world.** Both one's and others' actions shape the external world as an outcome. Self-world interaction follows a partially observable Markov decision process (POMDP), while other-world interaction follows an inverse POMDP. That is, the agent does not fully know the world state and others' intentions. When another agent strongly influences the world, the agent prioritizes observing the other's actions (red) and observations (green) over the rest of the world state (blue).

more to others' actions to identify intentions relevant to their own problem-solving [11–13] (Fig 1, red arrow), especially when these actions are more socially valuable [14]. On the other hand, if observing others provides no particular benefits in learning, one may treat them as part of the uninformative surrounding environment [15,16] (Fig 1, blue arrow). Another example is social gaze–observing what others see (Fig 1, green arrow); more valuable information can be gained by observing how others selectively observe [17–20]. Therefore, the proposed framework is expected to integrate residual evidence from previous studies into a more coherent explanation by incorporating shifts in observational focus.

Within our framework, what would be the consequence of such selective observation in a social context? We reasoned that people update their inferences only when they gain new information from observation; otherwise, their inferences remain unchanged. This suggests that selective observation shapes inference by prioritizing certain social cues over others under a resource constraint. To test this, we designed an interactive task in which participants inferred the hidden intentions of a computer-controlled agent embedded in a dynamic social setting. The agent's behavior could unexpectedly shift from cooperative to competitive, or vice versa, eliciting positive or negative prediction errors. Because participants could not be monitored simultaneously, they had to allocate their gaze to resolve uncertainty strategically. This design allowed us to investigate how inference biases emerge from the interplay between past social experience and observational focus.

We show that participants' inferences about the opponent's intentions were shaped by the recent interaction history, particularly following betrayal. This history dependence was asymmetric: accumulated betrayal led participants to infer greater competitiveness than warranted, whereas comparable experiences of unexpected help failed to induce a similarly strong bias toward cooperative intent. We formalized this asymmetry using an energy landscape framework, revealing that repeated betrayal shifted the attractor basin of the inference dynamics toward competitive interpretations. Eye-tracking data further demonstrated that this shift was followed by a selective allocation of gaze toward the opponent at the expense of monitoring one's own avatar, reflecting a trade-off between information sampling and control. To probe the mechanism further, we trained a task-optimized recurrent neural network (RNN), which reproduced the asymmetrical inference pattern and revealed that redirecting observational focus could reverse its sensitivity to betrayal. These results suggest that selective observation under resource constraints amplifies path-dependent social inference, offering a unified framework that links attention, decision dynamics, and internal models of others.

PLOS Computational Biology

## Results

### Interactive social task to infer the hidden intention of the opponent

Sixty-two participants (35 males, 27 females; age: 23.3±2.9) played a two-dimensional interactive prey-pursuit game (Fig 2A). Using a joystick, they were required to catch prey that fled from both their avatar and a computerized opponent (Fig 2B). The reward amount decreased as the capture time increased. The opponent's moment-by-moment movement was determined by a hidden intention parameter (friendliness, F) that balances two conflicting goals: reducing its distance to the prey or the participant's distance to the prey (Fig 2C). A higher F led the opponent to reduce the distance between the participant and the prey, herding the prey closer to the player. In contrast, a lower F led the opponent to reduce the distance to itself, intercepting the prey before the player. Each trial started with participants watching, on average, a 2.3-second video of three characters interacting, and their eyes were allowed to move freely (inference phase). They were instructed to evaluate the opponent's hidden intention during the inference phase and then decide whether to increase (i.e., boost) or decrease (i.e., hinder) the opponent's speed in the subsequent pursuit phase. Strategic decisions were essential since the prey was set to be faster than the participant; boosting a competitive opponent increased the likelihood of failing to capture the prey within the time limit, whereas boosting a cooperative one increased the likelihood of a successful capture.

Unbeknownst to participants, the opponent's F value changed after their decision (Fig 2D). F value decreased during the pursuit phase compared to the inference phase in the betrayal scenario, whereas it increased in the unexpected-help scenario. In Experiment 1, these scenarios alternated probabilistically. In betrayal, the opponent appeared to herd the prey toward the player in the inference phase but intercepted it in the subsequent pursuit phase (Fig 2E, left panel; S1 Video). On the other hand, in unexpected help, the opponent chased the prey in the inference phase but herded it toward the player in the pursuit phase (Fig 2E, right panel; S2 Video). Participants acquired more monetary reward by boosting the opponent with high-F in the inference phase and hindering the opponent with low-F (Fig 2F; choice boost VS. hinder in low-F, mid-F, and high-F, $t_{47}$ = -3.243, -0.902, 9.322 with $p$ = 0.002, 0.372, 2.932e$^{-12}$, respectively). They earned lower rewards when mistakenly boosting a betraying opponent but gained more when boosting an unexpectedly helpful one (Fig 2G; betrayal and unexpected help, $t_{47}$ = -12.568, 20.105, with $p$ = 1.225e$^{-16}$, 1.007e$^{-24}$, respectively). These results demonstrate that the interactive social task prompted participants to distinguish between betrayal and unexpected help and adjust their choices accordingly to maximize rewards.

### Asymmetrical history dependence in inference

Humans learn from others' past actions and intentions, which in turn shape subsequent social inference [3, 21]. We therefore hypothesized that participants would infer the opponent's hidden intentions differently as a function of interaction history, even when observing behavior generated by the same F value. Consistent with this prediction, after experiencing betrayal, participants were less likely to boost the opponent than warranted by the opponent's actual F value (Fig 3A shows the across-participant average; S1A Fig shows individual subject data; red lines below the unity line; $t_{47}$ = -4.130, $p$ = 1.477e$^{-4}$). In contrast, after unexpected help, participants showed no systematic bias, and their choices aligned the true F values more accurately (blue lines overlap with the unity line; $t_{47}$ = 0.702, $p$ = 0.486). Eight participants in the present sample were recruited in a separate session to assess the robustness of this effect, and showed a comparable pattern of results (S1B Fig). Together, these results reveal an asymmetric history effect: betrayal shifted inferences toward a more competitive interpretation, whereas matched unexpected help produced little change. This asymmetry was independent of sex (S1C Fig; for male, $t_{26,betrayal}$ = -3.133, $p$ = 0.004, $t_{26,unexp.\ help}$ = -0.674, $p$ = 0.506; for female, $t_{20,betrayal}$ = -3.014, $p$ = 0.007, $t_{20,unexp.\ help}$ = 1.807, $p$ = 0.086) and age (S1D Fig; $\rho_{46,betrayal}$ = -0.067, $p$ = 0.653, $\rho_{46,unexp.\ help}$ = 0.014, $p$ = 0.927). The magnitude of the effect increased with continued betrayal: repeated betrayal progressively reinforced the bias toward more competitive inferences (S1E Fig; linear trend analysis, $F_{(17, 47),\ betrayal}$ = 17.863, $p$ = 2.648e$^{-5}$; $F_{(17, 47),\ unexp.\ help}$ = 4.901, $p$ = 0.027). However,

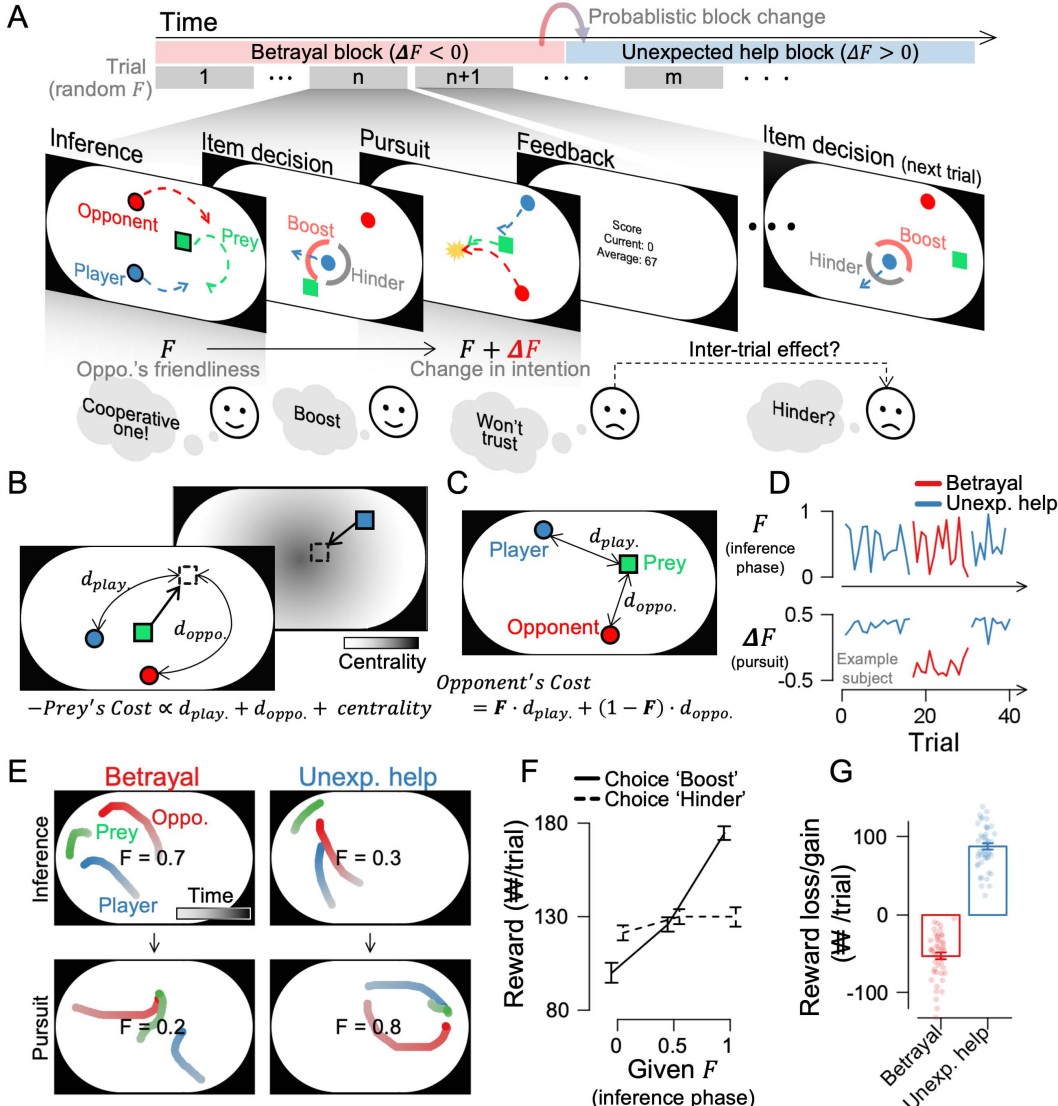

**Fig 2. Task schematics of Experiment 1. (A)** Participants watched videos of their avatar (green) and a computer opponent (red) pursuing prey (blue). The opponent's hidden intention (F) ranged from competitive (F=0) to cooperative (F=1). Participants then decided whether to boost or hinder the opponent's speed before pursuing prey themselves with a joystick. Unbeknownst to the participants, the F value decreased during the pursuit phase in the betrayal condition and increased in the unexpected help condition. Participants experienced each condition in blocks, with random changes occurring about every 20 trials. **(B)** Prey algorithm: The prey fled from both agents to maximize distance, avoiding walls via a centrality cost. **(C)** Opponent algorithm: At F=1, opponents minimized the distance between the player and the prey (cooperative); at F=0, they minimized the distance between themselves and the prey (competitive). **(D)** In the inference phase, we set the F value randomly on each trial. In the pursuit phase, we shifted this value upward or downward across blocks to induce an unexpected-help or betrayal condition. **(E)** Example trials. Left: In the betrayal condition, the opponent herded the prey during the inference phase and intercepted it during the pursuit. Right: In the unexpected help condition, the opponent intercepted the prey during the inference phase and herded it during the pursuit. See also S1 and S2 Videos. **(F)** Boosting a high-F opponent was relatively more rewarding, whereas hindering a low-F opponent was relatively more advantageous.**(G)** Reward losses from boosting a betraying opponent and gains from boosting an unexpectedly helpful opponent. Error bars represent ±1 SEM, and dots indicate individual participants. ₩ denotes the Korean currency, won.

PLOS Computational Biology

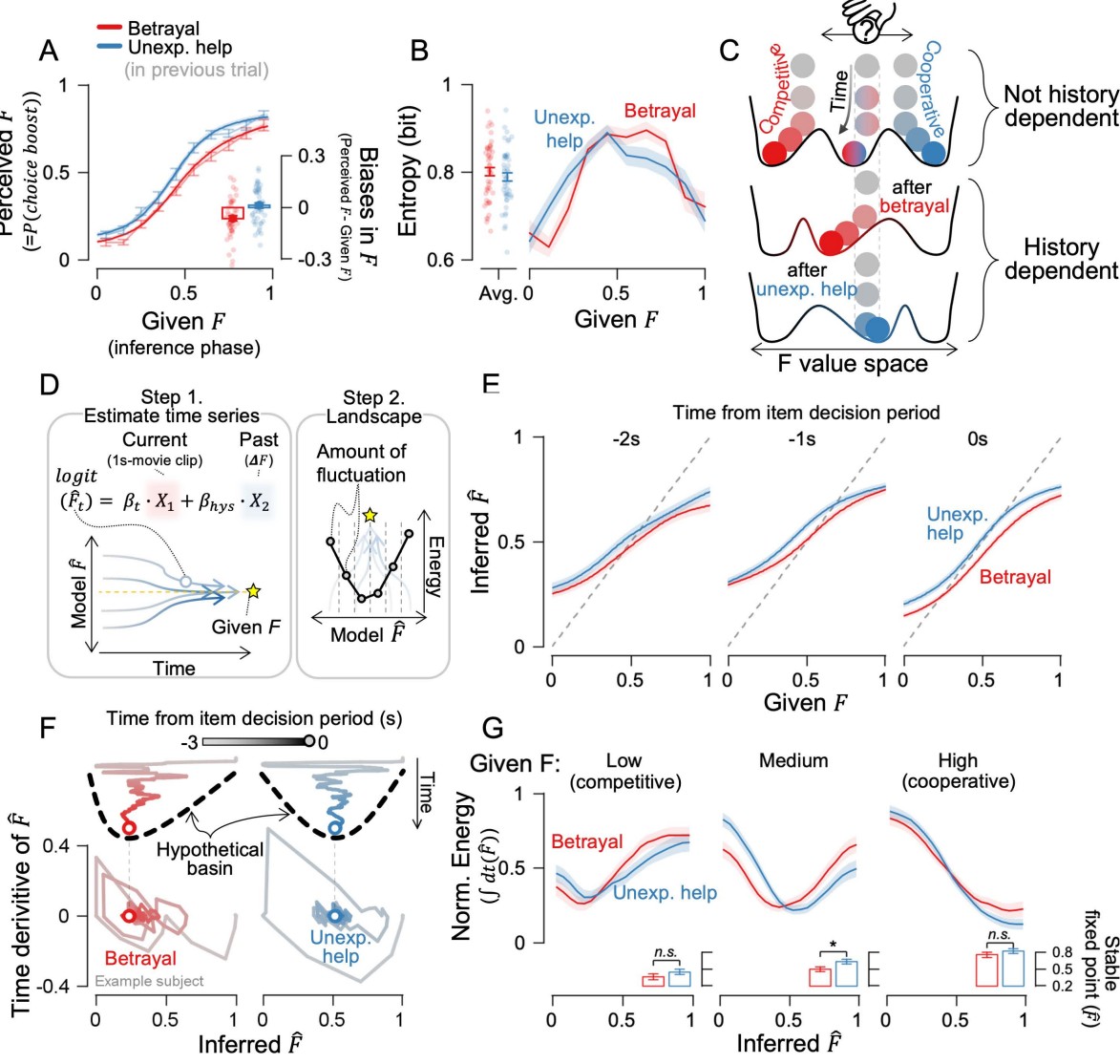

**Fig 3. Asymmetrical history dependence in inference. (A)** The psychometric curve shows the probability of choosing the boost item dropped after betrayal. Bold lines show cumulative Gaussian fits, and faint lines show binned data. The subfigure shows biases in their decisions, calculated as perceived F minus given **F. (B)** Shannon entropy of given F showing uncertainty in participants' choice. **(C)** F-value inference during the inference phase can be represented as a ball rolling on a landscape with basins. If prior history alters the landscape, identical stimuli would settle at different points. **(D)** Schematic of inference landscape reconstruction: moving-windowed trajectories during the inference phase and the previous trial's ΔF were used in an autoregressive logistic regression model. Temporal changes in model-estimated F̃ were accumulated to build the energy landscape. **(E)** The psychometric curve, derived from the logistic regression model, shows how inference evolves over time until the decision phase begins. **(F)** Example subject's phase portrait of inferred F̃ value after betrayal and unexpected help. A positive F̃ derivative on the y-axis indicates an increase in F̃ at the next time point on the x-axis, and vice versa. **(G)** Reconstructed energy landscapes, split into trials with low (F < 0.33), medium (0.33 < F < 0.67), and high F value (F > 0.67). The lowest energy points (stable fixed points) show where inferences would stabilize in the subfigures. Shaded ribbons and error bars represent ±1 SEM, and stars indicate statistical significance (*, p < 0.05; n.s., not significant).

the biases after the betrayal experience immediately returned to a neutral state when the betrayal ceased (i.e., at block change) (S1F Fig). This suggests that asymmetry accumulates and strengthens with repeated experiences of betrayal, whereas inference can be rapidly updated to the changed context when betrayal is abruptly removed.

Given the asymmetrical history dependence in social inference, we conducted a comprehensive evaluation of alternative factors that could account for this effect. One possibility is that betrayal caused greater monetary loss than the gain from unexpected help. However, when losses and gains were computed as the difference in reward between boost and hinder decisions within each condition, we observed the opposite pattern: the gain was larger in the unexpected help condition (Fig 2G; betrayal VS. unexpected help, $t_{47}=6.276$, $p=1.028e^{-7}$). This indicates that reward value differences do not drive the asymmetry. Other factors, such as loss aversion [22], could also contribute to asymmetries in intention inference. However, perceived F values on the next trial did not vary with reward magnitude on the previous trial (S2A Fig; slope, $t_{47}=1.237$, $p=0.222$).The same result held when we compared the probability of choosing the boost item on the next trial following smaller versus larger rewards on the previous trial (S2B Fig; lower- vs higher-reward, $t_{47}=-0.802$, $p=0.427$). We also considered whether participants strategically manipulated their reports. If so, such manipulation should be least feasible when perceptual evidence is near threshold. Yet even when the change in $F$ was within half of the just noticeable difference (JND), responses still differed reliably between betrayal and unexpected-help conditions (S2C Fig; betrayal VS unexpected-help, $t_{47}=4.208$, $p=1.150e^{-4}$), arguing against a conscious-response strategy as the primary driver of the asymmetry. We next asked whether the opponent's observable behavior differed across conditions on the subsequent trial, which could provide an external cue for differential updating. Basic kinematic and geometric features, including agents' positions, speed, acceleration, and angle relationships, were indistinguishable following betrayal versus unexpected help (S3 Fig). Thus, participants arrived at different decisions despite closely matched sensory evidence, suggesting that history primarily altered internal computations rather than the stimulus statistics.

As another possibility, we hypothesized that prior interaction history would influence the internal process in the next trial, particularly by increasing uncertainty when the opponent displayed ambiguous intentions. We converted choice probabilities into Shannon entropy (see Materials and methods for details), a measure used for quantifying the internal uncertainty of participants choosing certain options [23–25]. Mean entropy did not differ between betrayal and unexpected-help trials (Fig 3B, left panel; $t_{47}=1.031$ with $p=0.308$). However, when entropy was analyzed at each level of the given F value separately, participants' confidence in their choices varied according to their interaction history (Fig 3B, right panel); the entropy was higher following betrayal than unexpected help when $F<0.5$, but this pattern reversed when $F>0.5$, with higher entropy following unexpected help (Cluter-based permutation t-test, betrayal<unexpected help, $t_{47}=2.424$, 2.468, with $p=0.002$ for both, at $F=0.15$ and 0.25, respectively; betrayal>unexpected help, $t_{47}=-3.100$, -2.159 with $p=0.002$ at $F=0.65$ and 0.75, respectively). Additionally, decision times were similar across conditions (S4A Fig; betrayal vs. unexpected help, all $p>0.05$), and they were not associated with the magnitude of the inferential biases (S4B Fig; $t_{47}=1.490$, $p=0.143$), indicating that entropy differences reflected shifts in internal uncertainty (and where certainty peaked) rather than uncertainty arising from longer decision times. Together, our results carefully examined the source of asymmetric history dependence: it is not explained by reward magnitude, perceptual/kinematic differences, or deliberate response manipulation, but instead reflects history-dependent changes in the internal uncertainty landscape over intention inference, expressed selectively across the evidence axis ($F$).

## The energy landscape captures the asymmetry

Uncertainty in participants' choices relates to history-dependent inference, but how this uncertainty translates into inferential bias remains unclear. We approached this question using the framework often referred to as an energy landscape and interpreted uncertainty reduction as the system settling into an energetically stable state (i.e., stable fixed point) [25] (Fig 3C). The energy landscape framework transforms uncertainty into an intuitive geometric representation and offers predictive power, as decisions follow subject-specific dynamics shaped by uniquely formed landscapes.

We reconstructed the energy landscape by tracking how inference stabilized over time using an autoregressive logistic regression model (Fig 3D). This model captured how participants inferred the opponent's intention moment by moment from the temporal patterns observed in the movements of the three characters. Specifically, during the final

1-second inference window, we computed the pairwise Euclidean distances among the three characters and averaged them in 200-ms bins, yielding 15 representative features that served as regressors. Thus, the 15 coefficients for this regressor, $\beta_t$, reflect how accurately participants inferred the opponent's intention from the trajectories of the three agents during the inference phase, and weakening $\beta_t$ correspondingly attenuates this inference sensitivity (S5 Fig, compare the differences across rows). We also included the strength of betrayal or unexpected help experienced in the previous trial ($\Delta F_{previous}$) as an additional regressor. The coefficient for this regressor, $\beta_{hys}$, indicates how strongly past experiences shaped the magnitude of the resulting inference bias (S5 Fig, compare the differences across columns). S1 Table shows the fitted results of the regression model. By applying this procedure frame by frame for each trial, the model produced an inferred time series showing how participants' estimates evolved toward their final decision for a given inference-phase movie clip. When the inferred time series converges toward the given F value over time, little change is observed near the center of the F range, whereas large fluctuations appear toward the extremes. To quantify this pattern, we computed the time derivative of the inference time series within each F-value bin and integrated it to estimate the degree of instability [25]. Based on this instability measure, we then reconstructed the inference landscape (see Materials and methods for details).

We first examined when history-dependent inference arises—whether it stems from initial conditions before inference begins, or from dynamic changes as inference unfolds. The model supported the latter hypothesis. Asymmetric biases gradually emerged in the psychometric curves, with betrayal leading to a lower F value compared to unexpected help (Fig 3E), mirroring the final decision pattern shown in Fig 3A. Moment-by-moment changes in inference—known as a phase portrait—visualized this more clearly; while early fluctuations appeared similar, inference eventually settled down to a more competitive interpretation after betrayal (Fig 3F). This suggests that asymmetrical history dependency arose from changes in inference dynamics rather than from an initial baseline condition shift.

Which aspect of the inference dynamics changed after betrayal? We considered two possibilities. First, the convergence point becomes more deeply embedded after betrayal than before, reflecting a higher degree of confidence in the decision. This resembles a steeper or deeper valley—harder to escape and more stable—which is known to reflect stronger decision confidence [25–27]. Second, the convergence point itself may shift laterally, meaning the bottom of the valley moves. The reconstructed energy landscape supported the convergence shift, especially when the opponents behaved ambiguously. The points of lowest energy shifted significantly toward lower inferred F values after betrayal, especially when participants encountered an opponent displaying ambiguous behavior with mid-range F values (Fig 3G; betrayal vs. unexpected help across low-F, mid-F, and high-F; $t_{47} = 2.185, 2.433, 1.341$; $p = 0.034$, 0.019, 0.186, respectively). In contrast, the steepness or depth of attractor basins remained consistent across different prior experiences (S6A Fig; across F-value ranges, $F_{(2,94)} = 8.026, 0.450$, with $p = 4.519e^{-4}$, 0.638; across betrayal VS. unexpected help, $F_{(1,94)} = 0.014, 0.395$, with $p = 0.907$, 0.531; and interaction, $F_{(2,94)} = 0.522, 0.014$, with $p = 0.594$, 0.986, depth and steepness, respectively). The result indicates that history-dependent inference arises not from changes in the degree of confidence (i.e., deeper basin), but rather from the shift in decision criteria (i.e., shift of basin).

Because basin geometry and basin position have direct behavioral analogs, with basin depth and steepness relating to sensitivity ($d'$) and basin position relating to decision criterion, we next tested these landscape-derived predictions using Signal Detection Theory. There were no differences in the sensitivity across conditions (S6B Fig; betrayal VS. unexpected help, $t_{47} = 1.827$, with $p = 0.074$). Instead, the decision criterion differed markedly across conditions, remaining near zero after unexpected help but shifting to a significantly positive value after betrayal (S6C Fig; betrayal: $t_{47} = 3.336$, $p = 0.002$; unexpected-help: $t_{47} = 0.083$, $p = 0.934$; betrayal vs. unexpected-help: $t_{47} = -4.070$, $p = 1.788e^{-4}$). This result corresponds to the energy landscape results, indicating that after betrayal, participants adopted a more conservative criterion and required clearer cooperative behavior before selecting an option to boost the opponent.

## Observational focus shifts adjust inference

We next asked how prior experience shifts the endpoint of inference. One plausible mechanism is biased evidence sampling, in which inferential bias emerges because participants preferentially acquire negative evidence on some trials. Because the inference phase provides the most direct informational basis for the upcoming decision and imposes minimal motor demands, we first examined gaze allocation during this phase. And we found that gaze patterns were highly similar across conditions (Fig 4A). A ternary plot of gaze allocation during the inference phase showed that the mean gaze location was centered among the three agents (Fig 4B, top). When the analysis was restricted to a direct comparison between the player's avatar and the opponent, the relative distance between gaze location and each agent likewise did not differ across conditions (Fig 4B, bottom; cluster-based permutation test, betrayal vs. unexpected help, all time points p > 0.05). By contrast, gaze during the preceding trial's pursuit phase diverged markedly while participants experienced betrayal and unexpected help in real time (Fig 4C). Gaze differed across conditions in whether it was biased toward the player or the opponent, whereas the extent of prey-directed gaze did not differ (Fig 4D, top). When gaze was examined over time with respect to the opponent versus the player, participants shifted their gaze toward the opponent only in betrayal trials, whereas no such shift was observed in unexpected-help trials. (Fig 4D, bottom; overall trials, p < 0.001 after 1.10 seconds

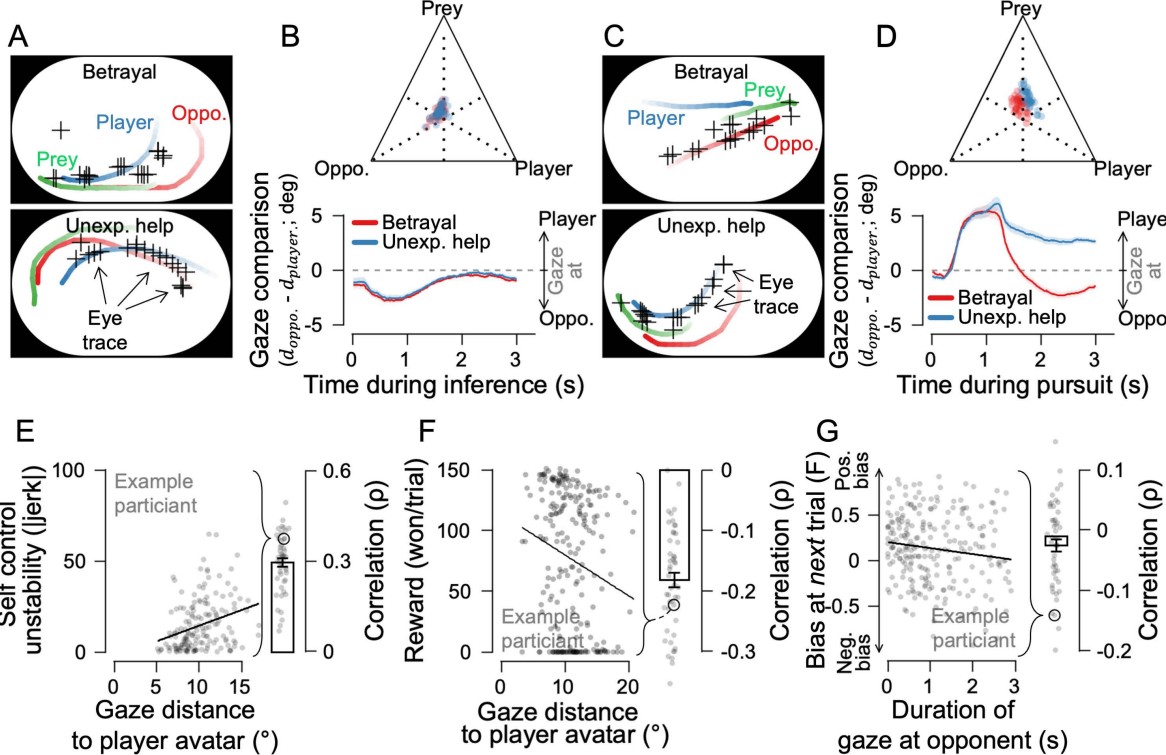

**Fig 4. Observational focus shifts adjust inference. (A)** Eye-tracking examples show that the participant mainly observed the opponent in the inference phase, regardless of previous experience. **(B)** Top: a ternary plot of gaze allocation across the three agents during the inference phase. Bottom: comparison of the distance from the gaze center to the opponent's avatar ($d_{oppo.}$) and the player's avatar ($d_{play.}$) during the inference phase. **(C)** Eye-tracking examples show gaze shifts toward the opponent only during betrayal trials but not during unexpected-help trials in the pursuit phase. **(D)** Top: A ternary plot of gaze allocation across the three agents during the pursuit phase. Bottom: gaze comparison reveals a systematic shift in observational focus while experiencing betrayal in the pursuit phase. **(E)** A farther gaze from oneself correlates with lower stability (higher jerk) in the self's avatar movement. **(F)** The farther participants looked from their own avatar, the less reward they earned. **(G)** Longer gaze durations toward the opponent correlate with biases in perceived F in the subsequent trial. Error bars and shaded ribbons indicate ±1 SEM, with dots representing individual trials.

from the pursuit phase onset; S7A Fig, when F values matched, p < 0.001 after 1.17 seconds from the pursuit phase onset). As with S1B Fig, eight participants in the present sample were recruited in a separate session and showed the same gaze-shift pattern toward the opponent in betrayal trials but not in unexpected-help trials (S7B Fig).

Why did participants look at the opponent? During the pursuit phase, participants needed to precisely control their own avatar to catch the prey, not monitor the opponent's movements. In fact, the more they looked away from their own avatar, the less stable their movement became (Fig 4E; $t_{47} = 27.772$, $p = 6.200e^{-26}$), and rewards decreased accordingly (Fig 4F; $t_{47} = -14.783$, $p = 2.701e^{-19}$). Despite this cost, participants continued to shift their gaze toward the opponent, and we found that the longer they looked at the opponent, the more likely they were to infer competitive intent (Fig 4G; $t_{47} = -2.481$, $p = 0.017$). Longer viewing durations produced the comparable effect (S7C Fig; $\rho = 0.032 \pm 0.008$, mean ± SEM; $p = 0.010$ from 1.25–1.80 s and $p = 0.001$ from 1.83–3 s after pursuit onset). Together, these results suggest that, despite a measurable cost to motor control, participants prioritized monitoring the opponent's betraying behavior. By selectively observing the opponent, participants may have gained disproportionate information about competitive behavior, thereby contributing to an information asymmetry that biases subsequent inference.

## Observation target change counteracts inferential bias

Does inferential bias change when the target of observation is altered? Because task-optimized RNNs allow us to precisely manipulate the information available to the inference process while holding the task structure fixed, we trained an RNN on the same environment and then perturbed its observational inputs to test whether selective changes in what is observed causally shift inferred intention. (Fig 5A). As a first step, we trained an RNN to predict a scalar F value from inference-phase movies without using participant choices, yielding a task-optimized RNN that served as a neutral inference machine. Next, we asked how this network adapts when it learns from pursuit-phase movies in which participants experienced either betrayal or unexpected-help scenarios. When we fed these pursuit movies into the neutral inference machine from the first step, the inferred value F̃ diverged systematically from the expected F (defined from the inference phase): betrayal scenarios produced negative deviations, whereas unexpected-help scenarios produced positive deviations. To quantify betrayal sensitivity, we introduced a free parameter, $\beta_{bet}$, that controls the relative weighting of learning from betrayal versus unexpected help. During this adaptation, the network also updated the input weights associated with the three agents, and we treated these weight changes as a proxy for selective observation in humans. Using the network from the first step as a fixed baseline, we repeated the adaptation as the second step to characterize how distinct learning histories drive divergence in the network's subsequent inferences.

Simulations showed that the inference networks bifurcated into two categories, becoming selectively sensitive to either betrayal or unexpected help (Fig 5B). This bifurcation arises because when $\beta_{bet}$ deviates above or below 0.5, the model becomes more sensitive to one of the two conditions, and repeated training epochs amplify this imbalance toward one side. The distributions of input F values used for training were comparable between betrayal-sensitive and unexpected-help-sensitive networks (Fig 5C). However, betrayal-sensitive networks produced negatively biased inferred F values, whereas unexpected-help-sensitive networks showed no such negative bias (Fig 5D). This asymmetry mirrors the history-dependent bias observed in human participants following betrayal versus unexpected help (Fig 3A, inset).

Why does bias emerge selectively when the network learns from betrayal? In our task, the latent variable F parametrizes the opponent's intention, so the most diagnostic evidence for F should reside in the opponent's movements, not in the player's movement. This means that players can learn negative inferential bias from a betraying opponent's movements and, in principle, positive bias from an unexpectedly helpful opponent. However, because selective observation of the opponent increases only in betrayal contexts, bias learning becomes asymmetric: observing betrayal strengthens negative bias, whereas overlooking unexpected help prevents positive bias from forming. Consistent with this prediction and with participants' observation patterns, the input-weight updates from step 1 to step 2 showed that betrayal-sensitive networks selectively strengthened opponent-related weights, whereas unexpected-help-sensitive networks

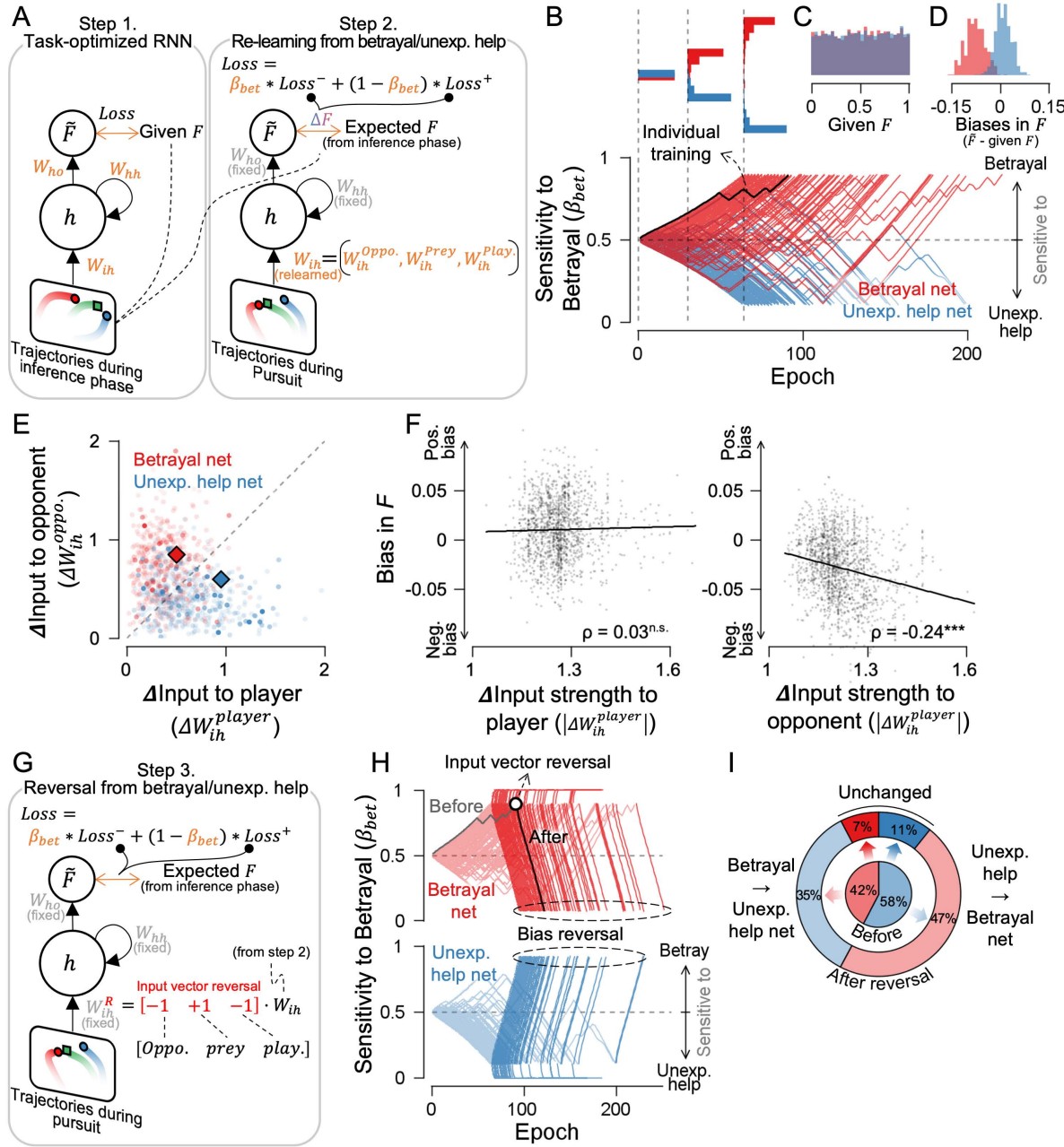

**Fig 5. A task-optimized recurrent neural network (RNN) demonstrated that observation can shape inferential bias. (A)** RNN training. Step 1: Train the model to predict the F value from the trajectory movie during the inference phase. Step 2: Re-train the model using the movie during the pursuit phase under betrayal or unexpected help conditions. Only the input weights $W_{ih}$ for the opponent, prey, and player were updated while keeping other parameters fixed. $\beta_{bet}$ balances sensitivity to betrayal and unexpected help. Orange indicates trainable parameters. **(B)** $\beta_{bet}$ over training epochs in Step 2. Each line represents a separate re-learning from Step 1, categorized as either betrayal-sensitive or unexpected-help-sensitive if $\beta_{bet}$ exceeded 0.9 or dropped below 0.1. The upper subfigures show diverging distributions over time. **(C)** Distributions of input F value. **(D)** Output biases in inferred $\tilde{F}$ value. **(E)** Absolute value of input weight changes for opponent and player movement trajectories. **(F)** Bias in F showed no correlation with changes in input to the player unit but showed a significant negative correlation with changes in input to the opponent unit. **(G)** Step 3: The input weights learned in Step 2 were sign-reversed, and only the betrayal sensitivity parameter $\beta_{bet}$ was re-trained to test the effect. **(H)** Learning traces of $\beta_{bet}$ before (Step 2, faint lines) and after sign reversal (Step 3, bold lines). **(I)** Most betrayal-sensitive networks switched to unexpected-help-sensitive networks, and vice versa, after reversal.

primarily strengthened player-related weights (Fig 5E). Crucially, across simulations, the magnitude of opponent-related weight changes (vector norms) tracked the degree of negative bias in inferred F, whereas comparable changes in player-related weights were largely unrelated to inferential bias (Fig 5F; $\rho = 0.034$ for player weights with $p > 0.05$ and $\rho = -0.240$ for opponent weights, $p = 3.762 \times 10^{-21}$). Together, these results suggest that betrayal-driven learning biases inference by concentrating learning on the opponent channel, which carries the most informative signals about latent intent in competitive contexts. As a result, competitive opponent kinematics exert a disproportionate influence on inferred F, whereas learning that prioritizes the player channel primarily improves control without systematically shifting the inferred intention.

Given that inference depends on observation, we wondered whether shifting observational focus alone could significantly alter it. We therefore reversed the opponent and player weights—serving as a proxy for observational input—in the task-optimized RNN, and examined changes in betrayal sensitivity ($\beta_{bet}$), while keeping all other parameters fixed. (Fig 5G). We found that reversing the signs of input weights, without altering the network structure, was sufficient to reverse sensitivity. Networks previously sensitive to betrayal became sensitive to unexpected help, and vice versa (Fig 5H), indicating that bias reversal occurred frequently across networks (Fig 5I). These reversal results support the idea that reallocating observational resources between the opponent and the self can alter inference in humans, and further suggest that such bias can be attenuated when the observational resources are redistributed.

## Hysteresis in social inference

We wondered whether inference would immediately become unbiased once betrayal stopped, or whether it would resemble hysteresis in physical systems—exhibiting lingering effects shaped by the trajectory of past experiences (Fig 6A). We assumed that if the amount of betrayal of the current trial's one has increased than the previous trial (i.e., more negative ΔF), participants are more likely to interpret the opponent as competitive, compared to when the error has decreased. Hysteresis rejects three alternative possibilities (Fig 6B): (1) inferential bias is independent of recent experiences; (2) all recent experiences have an equal and symmetrical effect on bias; and (3) even if asymmetrical, there is no path dependence—meaning no distinction between increasing and decreasing phases of betrayal. To test for hysteresis in inference, in the subsequent experiment (Experiment 2), we introduced a monotonic increase or decrease in the opponent's intention change (ΔF) between the inference and pursuit phases, alternating these phases to create a zigzag pattern of ΔF (Fig 6C).

As in Experiment 1, stronger betrayal experiences in previous trials led to a greater bias toward perceiving the opponent as a competitor, while unexpected help did not create a significant bias (Fig 6D-6E). However, this effect became more pronounced when the degree of betrayal increased across trials, compared to when it decreased (increase VS. decrease; $t_{13}$ = -2.347, -3.677, -3.739, -5.722, -6.713, -3.879, -5.180, with p = 0.035, 0.003, 0.003, 7.019e⁻⁵, 1.439e⁻⁵, 0.002, 1.768e⁻⁴, respectively for the seven bins ranging from the strongest to the weakest betrayal). A similar pattern of gaze behavior was observed, consistent with the findings from Experiment 1 (Fig 6F-6G). Participants generally focused on the opponent during betrayal and their own avatar during unexpected help, and the relationship between observation and inferential bias followed a similar trend as in the experiment 1 (S7D Fig). However, fixated on the opponent more closely during the phase of betrayal increase than during the phase of betrayal decrease (increase VS. decrease; $t_{13}$ = -3.042, -4.889, -4.644, -4.604, -7.836, -3.466, -2.122, with $p$ = 0.009, 2.957e⁻⁴, 4.592e⁻⁴, 4.934e⁻⁴, 2.801e⁻⁶, 0.004, 0.053, respectively for the seven bins ranging from the strongest to the weakest betrayal). This presents an interesting contrast, showing that even with the same experience of an opponent behaving with ambiguous intention, its impact on observation and subsequent inference differs depending on whether the degree of betrayal increased or decreased.

Hysteresis manifested as structural changes in the inference landscape, depending on whether the degree of betrayal was rising or falling. When we reconstructed the landscape from trials where the opponent's intention was ambiguous (i.e., mid-F range; see S2 Table for model fitting results), the stable fixed points with the lowest energy significantly differed

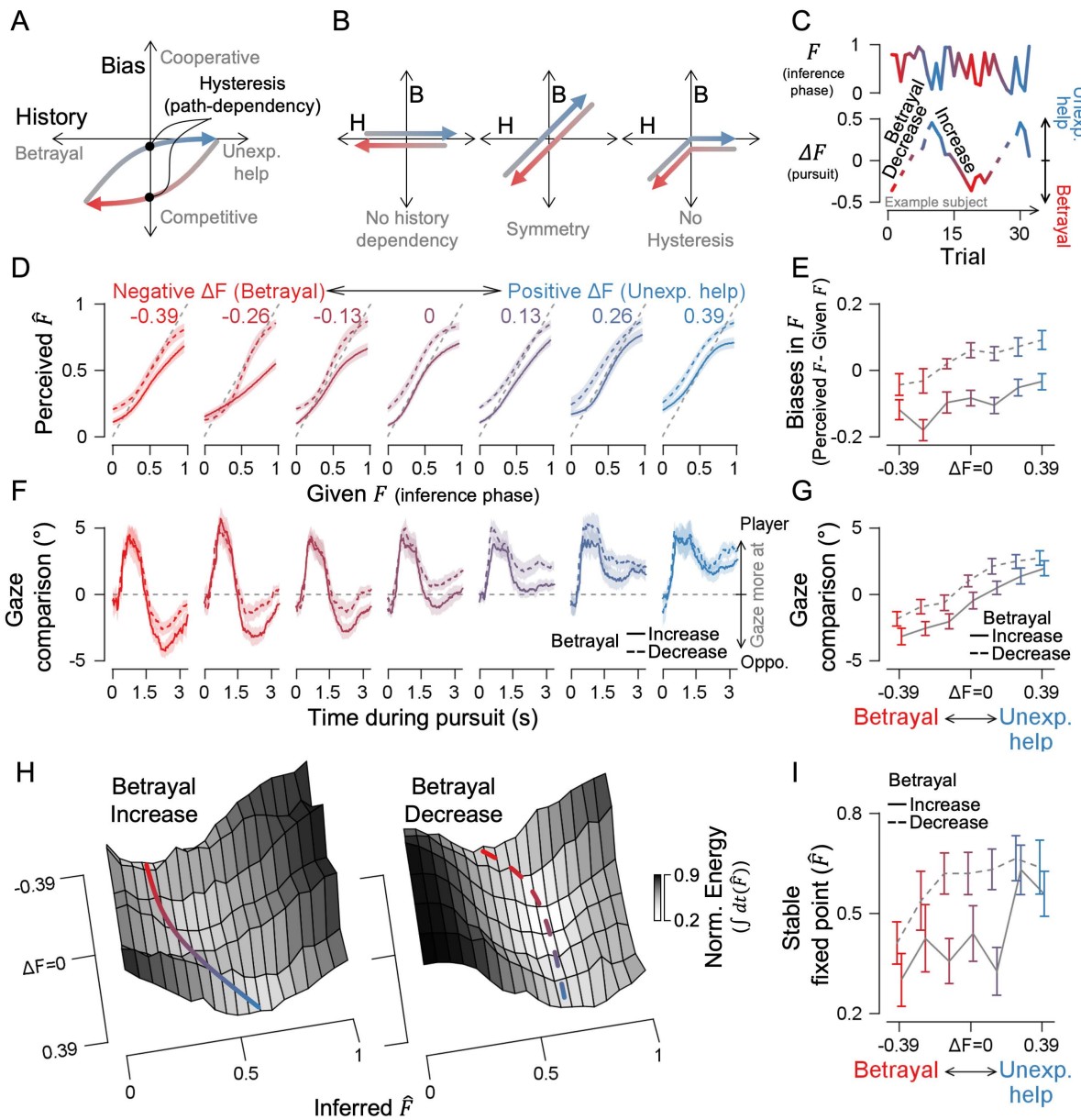

**Fig 6. Hysteresis in inference in Experiment 2.** **(A)** Null hypothesis: If the order of experience influences inference (i.e., hysteresis exists), a gradual increase and decrease in betrayal strength should bias inference differently. **(B)** Alternative hypotheses. Left: biases are history-independent; Middle: Biases depend on past experiences and each experience holds equal, symmetrical value; Right: biases depend asymmetrically on betrayal but not on the exact order of events (i.e., not path-dependent). **(C)** During the inference phase, the F value randomized while ΔF alternated in a zig-zag pattern, shifting between increasing and decreasing betrayal. **(D-E)** Psychometric curves and inferential biases by levels of betrayal or help. **(F-G)** Gaze comparisons between the player and the opponent by levels of betrayal or help. **(H)** Energy landscapes for moderate F value (0.33 < F < 0.67); basin lines mark the approximate lowest points. **(I)** Phase diagram showing where inference settles into lower-energy stable points. Shaded ribbons and error bars indicate ±1 SEM. Solid lines show an increasing betrayal phase, and dashed lines decreasing betrayal phase.

between the phases of betrayal increase and decrease, especially when ΔF was nearly zero at the latest trial (Fig 6H-6I; betrayal increasing VS. decreasing; $t_{13} = -2.337$ and $-2.623$, with $p = 0.036$ and $0.02$ for ΔF = $-0.13$ and $0.13$, respectively, while the rest showed $p > 0.05$). On the other hand, the inference landscape in trials with low or high F-values did not differ

between increasing and decreasing phases of betrayal (S9 Fig; only at $\Delta F = 0.26$ and high-F, $t_{13} = -2.503$ with $p = 0.026$, while the rest showed $p > 0.05$). This suggests that when the degree of betrayal continues to increase, inference does not immediately return to a neutral state due to its momentum, even if betrayal gradually subsides. Taken with Experiment 1, the present findings provide a more comprehensive account of the hysteresis-like behavior: its expression depends on the sharpness of the transition from betrayal to unexpected help, which in turn redistributes observational resources.

## Discussion

Like other forms of inference, inferring others' intentions is subject to bias, particularly as interactions unfold over time in partially observable environments. Here, we demonstrate such biases using an interactive prey-pursuit task involving an opponent with hidden intentions, interpreted through the lens of an energy landscape framework. The observed bias exhibited several notable properties. First, it was history-dependent: past interactions influenced current inferences, with the most potent effects occurring under high uncertainty. Second, the history dependence was asymmetric—participants showed more substantial biases following betrayal than unexpected help. Third, it exhibited hysteresis: the effect of betrayal history was greater when betrayal was an increasing phase than when it was a decreasing phase. We found that how participants allocated their observational resources shaped this bias, while balancing the trade-off between acquiring information about the opponent and maintaining precise control of the self-avatar. Simulation using a task-optimized recurrent neural network further suggested that shifting the focus of observation can reverse inference biases.

What are the functions of observation in social inference? In many situations, uncertainty challenges effective decision-making, especially when insufficient sampling misaligns external conditions with internal models [28]. In social contexts, this challenge is amplified: individuals often lack direct access to others' internal states and must selectively decide what to observe or sample (Fig 1). Traditionally, such uncertainty in social inference is thought to be resolved through high-level cognitive processes—such as adopting others' perspectives or simulating their experiences [29, 30], or by planning to reduce future ambiguity [31]. However, our results suggest that shifting gaze serves as a simple yet effective way to reduce uncertainty during social inference. More importantly, we emphasize that observation is selective and thus costly. This raises the question of how to reallocate limited observational resources to reduce specific aspects of social uncertainty. We suggest that a simple shift of gaze determines how the limited observational budget is allocated across social cues, leaving unobserved aspects without resources. For instance, when you concentrate on reading the audience's polite smiles, you might miss their hand gestures that signal boredom.

While decision-making models from non-social domains have provided valuable starting points for understanding social inference, they may capture only certain aspects of complexity in our results. For instance, a drift-diffusion model (DDM) might account for the current asymmetrical inferential bias by accelerating threshold-crossing after betrayal [1, 32–34] (Fig 7A), while a Bayesian model might explain it by assigning stronger prior beliefs to betrayal than to unexpected help [35–38] (Fig 7B). The canonical form of both models predicts reduced variability after betrayal compared to unexpected help. However, our data show no such difference in decision time or choice variability (Fig 3B, S6A Fig). To address this discrepancy, we adopted a dynamical systems perspective. A geometric structure in the inference energy landscape intuitively explains why asymmetric bias emerges only when betrayal increases (Fig 7C). This framework also explains why competitor bias is difficult to reverse once it has formed. Reversing it demands either consistent, strongly cooperative behavior from the opponent (green arrow) or redirecting observation toward the self (purple arrow). By embedding the influence of prior states, the dynamical system framework reveals the temporally intricate and path-dependent nature of social cognition.

Another benefit of the dynamical systems perspective is its ability to better explain the complex temporal dynamics of social cognition. Previous models, such as reinforcement learning based on social prediction error [39–42], often assumed that humans update inferences uniformly from any experience, regardless of the temporal order of previous experiences. However, the same experience can lead to different inferences depending on the order of events—a pattern well captured by hysteresis in dynamical systems. For example, in psychiatry, traditional models struggle to explain why depressive

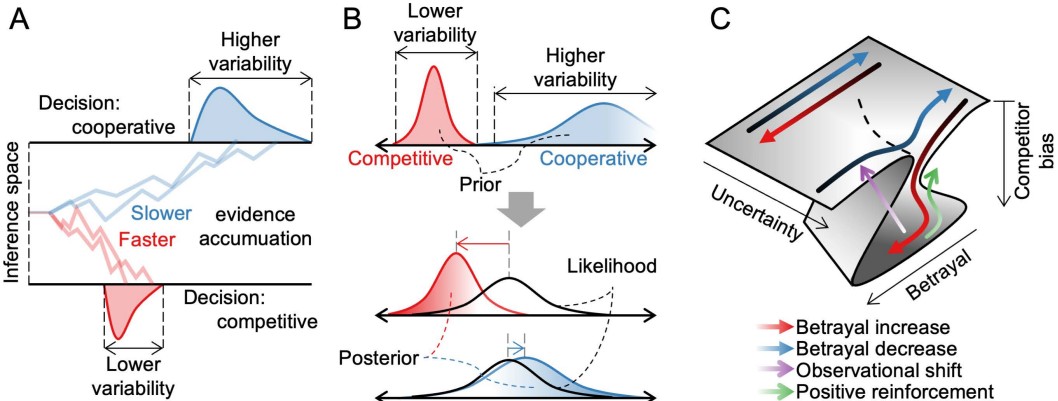

**Fig 7. The dynamical systems perspective and alternative viewpoints on hysteresis in social inference. (A)** Alternative view 1: Drift diffusion model (DDM), where social inference accumulates evidence over time until reaching cooperative or competitive decision thresholds. **(B)** Alternative view 2: Bayesian inference, where history sharpens the strong prior belief that the opponent is a competitor, biasing the posterior distribution. Both models predict less variable decision times or choices. **(C)** Dynamical system perspective: phase diagram showing where social inference stabilizes. In contexts with no uncertainty (extreme Fs), inference shows no bias regardless of betrayal. However, in ambiguous contexts (F ≈ 0.5), the increasing pattern of betrayal biases inference toward perceiving others as competitors, whereas a decreasing pattern does not. Recovery from bias can occur either by gradually reducing betrayal (green arrow) or shifting observation from others to oneself (purple arrow).

states persist even after stressors have subsided, whereas dynamical systems account for this persistence through hysteresis [43–46]. Hysteresis also illustrates inertia in neurological states that were previously unaccounted for [47, 48]. Similarly, we demonstrate hysteresis in social inference; humans continued to suspect betrayal even after it had ceased. Thus, in line with recent advances in viewing social interaction through the lens of dynamical systems [49, 50], we highlight the need to incorporate the temporal structure of experience into computational models to fully account for how social inference unfolds over time.

Although observation was emphasized as an active mechanism for social inference, this study is limited by the lack of direct manipulation of its influence. Similar to how occlusion in visual attention or first-person perspective constraints influence spatial navigation, direct manipulation of observability could shape social inference. For example, a study demonstrated that manipulating the gaze direction of computer agents altered observers' inferences about animacy [20]. More broadly, adjusting observability may help mitigate cognitive biases; individuals with paranoia and alexithymia often exhibit either excessive observation of others or reduced self-observability [51, 52], and individuals with autism spectrum disorder disproportionately observe non-social information [53–55]. Moreover, since observational focus shifts are often closely coupled with changes in attentional allocation, future work should also incorporate the role of attention. Therefore, future research should investigate whether manipulating observability influences inference and whether inference outcomes, in turn, shape subsequent observational choices. This can be reframed as a fundamental question of whether observational selection can function as a control mechanism for social inference within a closed-loop system. Though challenging, these questions prompt us to move beyond experimental designs that treat social inference as temporally discrete and independent and toward a more naturalistic and closed-loop setting [56–58].

## Materials and methods

### Ethics statement

The Institutional Review Board of Sungkyunkwan University approved the study (IRB 2022-11-009), and all studies followed relevant guidelines. All participants provided written informed consent and were naive to the purpose of the experiment. This study was not preregistered, and the analyses were exploratory and conducted after data collection.

## Participants

Data were collected from 62 human participants (35 male and 27 female; age = 23.3 ± 2.9). Forty-eight of them participated in Experiment 1 (27 male and 21 female; age = 23.3 ± 3.0), and the remaining fourteen were in Experiment 2 (8 male and 6 female; age = 23.1 ± 2.8). Eight participants in Experiment 1 were recruited in a separate session to assess the robustness of the effect.

## Experimental apparatus

All stimuli were displayed on a 24-inch LCD monitor (S27R350FHK) with a resolution of 1920 × 1080 pixels and a refresh rate of 60 Hz in a dark room. Participants rested their heads on a chin rest at a viewing distance of 60 cm from the monitor. They played the prey-pursuit game (see next section), controlled by a custom MATLAB (MathWorks Inc.) program using the Psychophysics Toolbox [59, 60]. They responded using a Logitech Extreme Pro 3D joystick. The horizontal and vertical eye positions of the right eye were simultaneously recorded using an EyeLink 1000 Plus infrared eye tracker (SR Research) at 1 kHz and later down-sampled to 60 Hz for post-hoc analysis. Eye tracker calibration was performed by presenting a circular dot (0.3° in diameter) at nine locations: the four corners, four sides, and the center of the screen. Calibration was repeated if the recorded eye position deviated visibly from the fixation point or if the participant lifted their chin from the chin rest.

## Prey-pursuit game task

Participants played a prey-pursuit game where they had to catch prey as quickly as possible while interacting with an opponent (Fig 2A). Each trial consisted of four phases. In the first phase (inference phase), they watched a video where three characters (player's avatar, computerized opponent's avatar, and prey) moved according to their pre-determined algorithms (see the sections below). The prey's initial position was randomly selected, while the player and opponent avatars were randomly placed within a range from 20° to 20.5°, maintaining an equal distance from the prey. The video ended automatically when either the player or the opponent came within Euclidean distance of 100 pixels of the prey, with an average duration of 2.3 s. This was to prevent the opponent or player from revealing hidden intentions too explicitly by catching the prey. The characters were outlined with a black border to prevent confusion with other phases. The video restarted from the beginning if their gaze deviated by more than 20° from the center of the three characters or if they tried to control the joystick. This was to ensure participants paid close attention during the inference phase. Participants were informed that the opponent could be either cooperative, assisting in the hunt, or competitive, intercepting the prey, but were not given details about their exact behavior patterns. The inference phase was introduced to allow participants to form an expectation about the opponent's role before making a decision. Separating this phase from the pursuit phase also prevents the inference process from being confounded by factors such as motor planning and execution.

When the item decision phase began, the black outlines of all three characters were removed, and their final positions from the previous inference phase were retained. At the same time, two colored arcs appeared around the player's avatar (3° apart). Participants were required to move the player's avatar toward one of the two arcs using a joystick (i.e., a two-alternative forced choice, 2AFC). Reaching the red arc made the opponent's speed faster in the subsequent phase (boost item), while reaching the gray arc slowed it down (hinder item). The angles at which the two options appeared were fully randomized. The item decision phase lasted until the participant responded, with an average response time of 1.3 seconds.

In the following pursuit phase, all three characters' initial positions were re-randomized using the same method for the inference phase. From the reset initial position, participants were instructed to control the player's avatar with a joystick to catch (touch) the prey's avatar. They had up to 15 seconds to catch the prey, with an average completion time of 7.6 seconds. If they succeeded within 2 seconds, their score in the feedback phase was 100 points and reduced proportionally (e.g., 100 points for catching the prey in 2 seconds, 50 points if caught in 8.5 seconds, and 0 points if not caught in 15 seconds). To prevent participant demotivation, if the opponent caught the prey before the player, the participant received a

small portion of the score they would have earned by catching it themselves. Specifically, the original score was multiplied by the opponent's F value and 0.5. For example, if the player had earned 80 points by catching the prey themselves, they received 8 points when the opponent's F value was 0.2 (8 = 80 × 0.2 × 0.5). The cumulative score was then proportionally scaled to determine their final experiment monetary compensation.

Participants were given a total of 20 minutes of pursuit time in Experiment 1 and 40 minutes in Experiment 2, excluding the inference phase and item decision periods. This time limit was set to encourage participants to catch the prey as quickly as possible. On average, participants did 216 ± 5 trials in experiment 1 and 505 ± 11 trials in experiment 2.

### Prey algorithm

The prey was designed to achieve two goals: escaping from the opponent and player avatars and staying near the center of the screen to avoid being cornered (Fig 2B). At each time point, the prey selected its next possible 15 locations, evenly spaced 24° apart in polar coordinates and equidistant in Euclidean space from its current position. If any of the 15 possible locations overlapped with a border or another character, that specific location was removed from the options. The change in Euclidean distances from the player and opponent avatars of each possible next position was summed, and min-max normalized across all 15 locations (escape cost), and the distance from the screen center was also min-max normalized (centrality cost). The final movement decision was made by weighing the escape and centrality costs at a 6:4 ratio and selecting the position with the lowest combined value. If the prey had visited the same location within the previous two frames, it did not select that position and instead chose the one with the lowest cost among the remaining options. When the opponent was within a distance of 25°, the prey fled at a maximum speed of 1.95°/s (130% of the player's default speed). If the opponent was farther away, the speed gradually decreased following a sigmoid function, approaching zero beyond 50°.

### Opponent's avatar algorithm

Similar to the prey's algorithm, the opponent selected its next position from 15 evenly distributed candidate locations at each time point. The candidate locations were ignored if they overlapped with borders or other characters. The opponent predicted the prey's subsequent position based on the prey's algorithm (see the above section) for each potential move. Then, the distances to the opponent's potential position ($d_{oppo.}$) and the current player's position ($d_{play.}$) were computed from the predicted subsequent prey position. The experimentally predetermined opponent's F value, ranging from 0 to 1, determined the opponent's cost for each potential position (Fig 2C), as follows:

$$Opponent's\ cost\ =\ F \cdot d_{play.}\ +\ (1 - F) \cdot d_{oppo.}$$

The opponent moves to the position with the lowest cost. This means that when F = 1, the opponent's movement would position the future prey closer to the player's avatar, whereas when F = 0, it moves to bring the future prey closer to itself. As a result, from the participant's perspective, F = 1 leads to a shepherding behavior, while F = 0 results in a prey-intercepting behavior (see Fig 2E and S1 and S2 Videos for examples). These modulations are known to be effective, as previous studies have shown that parametric modulation of physical features can influence perceived intention [61]. Note that while prior research indirectly modulated intention, we manipulated the computer agent's latent intention directly.

During the inference phase, the opponent's speed was 1.65°/s (110% of the player's default speed). During the pursuit period, if the participant selected the boost item in the item decision period, the speed increased to 2.13°/s (30% increase), whereas if they selected the hinder item, the speed decreased to 1.32°/s (20% decrease).

### Player's avatar algorithm

To make it difficult to catch the prey without the opponent's help, the prey's speed was 1.5°/s, 30% faster than the player's avatar's default speed. During the inference period, the player's avatar moved according to a predefined algorithm, which

predicted the prey's position 750 ms ahead based on its average movement direction over the last five frames and guided the avatar toward that location, with the default speed. This time window of prediction was determined based on previous research suggesting that the brain predicts approximately 750 ms into the future in similar pursuit tasks [62].

During the item decision and pursuit periods, the player's avatar moved proportionally to the joystick's force and direction, with a maximum speed when pushed at full strength. Vertical or horizontal joystick inputs that caused overlap with a border or other characters were ignored. To encourage accurate predictions of the opponent's F value during the item decision period, the player's speed in the pursuit period was slightly adjusted based on decision accuracy. Specifically, when the opponent's behavior was predictable (F = 1 or F = 0), and the choice was correct, the player's speed increased to match the prey's maximum speed. Conversely, if the choice was incorrect, the speed difference from the prey could increase up to twice the original gap. When the opponent's behavior was ambiguous (F = 0.5), the player's speed remained unchanged regardless of the participant's choice.

**Betrayal and unexpected-help conditioning**

A key manipulation in the experiment was that the opponent's F value during the inference period differed from the F value during the pursuit period without informing the participants. That is, this change occurred after participants decided whether to boost or hinder the opponent during the item decision period. If the change in F was negative, participants experienced a betrayal condition, whereas a positive change in F led to an unexpected help condition. From the participant's perspective, the opponent initially appeared to herd the prey toward the player in the betrayal condition but later intercepted it, whereas the opposite appeared to occur in the unexpected help condition. Example trials can be found in S1 and S2 Videos. In Experiment 1, the betrayal condition was repeated in a block design for 20 trials (with a random variation of ±4 trials) (Fig 2D). The block structure was used to allow participants to accumulate interaction history under a consistent violation pattern. To focus on expectation violations rather than gradual learning of the opponent's role, the F value was randomly assigned on each trial. The ΔF values were randomly set from −0.35 to −0.45 in the betrayal condition and from +0.35 to +0.45 in the unexpected-help condition. The value range was set so that most participants did not notice the change in *F* while still affecting their pursuit performance. In Experiment 2, the change in F value gradually increased from -0.45 to 0.45 over nine trials (betrayal decreasing phase) or decreased in the opposite direction (betrayal increasing phase) (Fig 6C). These two patterns alternated in a zigzag sequence of increases and decreases. In all cases, F remained within the range of 0–1.

**Analysis of item choice and reward**

To assess the relationship between participants' choices and monetary rewards, we binned F values during the inference period into three equal intervals and compared the average reward for each boost and hinder choice at the same trial (Fig 2F). To assess whether the betrayal and unexpected help conditions affected participants' monetary rewards, we also calculated the difference in overall rewards between boost and hinder choices (Fig 2G).

To examine whether prior experiences influenced choices, we binned the experimentally given *F* values from the inference period into ten equal intervals and calculated the probability of a boost choice by coding the boost choice as 1. Then, we fitted a cumulative Gaussian curve to capture the psychometric curve of intention inference. In Experiment 1, psychometric curves were computed separately for trials following the betrayal and unexpected help (Fig 3A for the across-participant average and S1A Fig for individual participants), and in Experiment 2, the change in F (ΔF) was binned into seven equal intervals (Fig 6D).

To quantify bias in inferences of the opponent's intention, we coded boost item choices as 1 and subtracted the given F value. Negative values indicated that participants rated the opponent's intention lower than its actual F value (competitive bias), while positive values indicated a higher rating (cooperative bias). The average bias in intention inferences for male and female participants separately is shown in S1C Fig, while the relationship between bias and age is presented

in S1D and S1E Fig presents the trial-averaged inferential bias and its changes as betrayal and unexpected help trials are repeated within the block. S1F Fig displays the same data realigned to the moment of block change, as the change occurred randomly around the 20th repetition. S4A shows decision times—measured from the onset of the item-decision phase to the moment the participant reached either the boost or hinder option—across conditions and across 10 bins of the given F value, and S4B shows the bias in perceived F across 10 bins of decision time (ranging from 0.6 s to 2.9 s in 0.2-s increments). In Experiment 2, the change in F ($\Delta$F) was divided into seven intervals, and inferential bias was calculated separately for each interval (Fig 6E).

To assess the sensitivity to loss, we also normalized trial-wise rewards to a 0–1 range for each participant and divided them into 20 equal bins, then measured perceived F (i.e., the probability of choosing the boost item on the next trial) within each bin (S2A Fig, left panel). For each participant, we quantified changes in perceived F (i.e., probability of choosing 'boost' item on the next trial) as a function of reward magnitude on the previous trial using the slope of a linear regression (S2A Fig, right panel). We then performed a median split on the previous trial's reward magnitude and compared slopes between lower-reward and higher-reward trials (S2B Fig).

We also examined whether participants might deliberately distort their responses by testing whether differences in perceived F emerged even within the just-noticeable-difference (JND) range (S2C Fig). JND was computed as half the difference between the 25% and 75% points of the fitted psychometric function, providing an estimate of the smallest detectable change in F. We computed perceived F within the JND range centered on the midpoint of the psychometric function (i.e., from the point of subjective equality).

To examine whether participants distorted their responses by detecting average physical properties of the two conditions, we computed the mean position, velocity, acceleration, pairwise distances, and angles among the three characters (S3 Fig).

To measure the internal uncertainty of participants' inference, the probability of choosing the boost item was converted into Shannon entropy (Fig 3B) as follows;

$$Entropy\ (bit)\ =\ -p(boost) \cdot log_2 p(boost)$$

When p(boost) is close to 0.5, entropy approaches its maximum value of 1 bit, indicating maximal uncertainty in choice. When p(boost) is close to 0 or 1, entropy approaches its minimum value of 0 bits, reflecting maximal certainty. Shannon entropy has been used as a measure of internal certainty [25], and more broadly, neural encoding of certainty is thought to manifest in choice probability [23, 24].

### Energy landscape analysis during the inference phase

To derive the structure governing the inference process, we built an autoregressive logistic regression model with two regressors (Fig 3D), where the regression equation was:

$$logit\left(P\left(boost_t\right)\right) =\ \beta_0\ +\ \beta_t \cdot X_t\ +\ \beta_{hys} \cdot \Delta F_{prev}\ +\ \epsilon$$

The first regressor, $\beta_t$, was on a short video clip capturing the trajectories of the three characters, $X_t$. To construct $X_t$, we computed the pairwise Euclidean distances among the three characters at each time point over a 1s period, then averaged them within 200-ms bins, resulting in 15 representative feature values (3 characters × 5 time points). The pairwise Euclidean distance was used to reduce confounding in the regression by focusing only on the relative spatial relationships among the prey, opponent, and player, regardless of their absolute positions on the screen. This approach was chosen specifically to avoid using raw coordinates, which could introduce irrelevant variability tied to the layout of the display.

The second regressor, $\beta_{hys,}$ was on $\Delta F_{prev}$, the scalar change in F in the previous trial (i.e., the amount of betrayal or unexpected help) experienced in the previous trial. Using these two regressors, we predicted whether participants would choose the boost item (coded as 1) or the hinder item (coded as 0) in the following item decision period. The model was trained separately for each participant using all data in experiment 1 from the inference phase, regardless of the betrayal or unexpected help condition. The same method was applied in Experiment 2, except that the betrayal-increasing and betrayal-decreasing phases were trained separately, based on their distinctive patterns shown in Fig 6D-6G. All model parameters were optimized through iterative maximum likelihood estimation.

Using the trained weights, we estimated the time series of inferred $\tilde{F}$ value for each trial. The inferred $\tilde{F}$ value represents the probability of the participant choosing the boost item, reflecting the strength of their judgment of the opponent as a cooperative helper. We then reconstructed the psychometric curve at each time point using the same method as in Fig 3A, based on the time series of inferred $\tilde{F}$ value for the betrayal and unexpected help conditions (Fig 3E). To analyze the meaning of each regressor on intention inference, we conducted simulations by multiplying 0, 1/3, 2/3, or 1 on $\beta_t$, or multiplying 0, 1, 2, or 3 on $\beta_{hys}$. We then estimated the psychometric curves based on the inferred $\tilde{F}$ value from the last 1 second before the item decision phase in the same manner as Fig 3E (S5 Fig). As $\beta_{hys}$ increases, the difference between unexpected help and betrayal conditions grows, while higher $\beta_t$ steepens the psychometric slope. This suggests that $\beta_{hys}$ underlies asymmetrical inferential bias while $\beta_t$ governs the inference process during the inference phase.

To reconstruct the energy landscape, we applied the principle that high-energy states exhibit greater temporal variation, while low-energy states show less variation over time. A similar approach is known to successfully reflect the decision space [25]. First, we divided the trials into three groups based on the given F value during the inference phase: low (F < 0.33), medium (0.33 < F < 0.67), and high (F > 0.67). It was further separated into the trials after betrayal and unexpected help in Experiment 1, and into seven equal bins of change in F value (ΔF) in Experiment 2. Second, time points were categorized into twenty equal intervals based on the inferred $\tilde{F}$ value at each moment, and the average time derivative was computed within each bin (as shown in Fig 3F). Third, the time derivative was integrated across the inferred $\tilde{F}$ value to estimate the energy landscape of inference. Specifically, we reconstructed the energy landscape U($\tilde{F}$) as

$$U\left(\hat{F}\right) \;=\; \int g\left(\hat{F}\right) d\hat{F}' \;\left(\text{where } g\left(\hat{F}\right) \;=\; \left|\frac{d\hat{F}}{dt}\right|_{\hat{F}}\right)$$

To determine the depth of the basin, we measured the range of energy values, from maximum to minimum (S6A Fig, left panel). Lastly, we performed min-max normalization on the energy landscape for a fair comparison across participants (Figs 3G, 6H, and S9A Fig). Assuming the lowest energy point as the stable fixed point of inference, we fitted first- to third-order polynomial functions to the energy landscape. We then calculated the Akaike information criterion (AIC) values for each fit and estimated the stable fixed point as the minimum of the polynomial function with the lowest AIC value (subfigures in Figs 3G, 6I and S9B Fig). The majority exhibited a U-shape, with a second-order polynomial function providing the best fit. To calculate the landscape's slope, we split the fitted polynomial function into two parts at the stable fixed point (the lowest point). We then applied linear regression to each part to compute the slope. The higher value of the two resulting slopes was defined as the basin's steepness (S6A Fig, right panel).

We additionally tested whether similar conclusions regarding basin depth and steepness could be obtained directly from the raw data by applying signal detection theory (S6B-S6C Fig). Specifically, we computed the sensitivity index (d′) and the decision criterion (c) as follows:

$$d' \;=\; Z(\text{hit rate}) - Z(\text{false alarm rate})$$

$$c \;=\; -\frac{1}{2}\left[Z(\text{hit rate}) \;+\; Z(\text{false alam rate})\right]$$

where Z is the inverse cumulative normal function.

## Gaze comparison analysis

To visualize how gaze was distributed across the three agents (player, opponent, and prey), we constructed a ternary scatter plot based on the relative proximity of gaze to each agent during the pursuit phase (Fig 4B and 4D, top panels). For each time sample, we computed the Euclidean distance between the gaze position and the screen position of each agent. We used the inverse of the average distance between the gaze location and each agent, computed either across the entire inference phase or within the 1.5–3.0 seconds window after pursuit onset. These inverse-distance values were then normalized so that the resulting weights summed to 1.

To compare whether participants focused more on the opponent or the player's avatar, we subtracted the distance from the gaze center to the screen location of the player's avatar from the distance to the screen location of the opponent at each time point. A negative value indicated that the gaze was closer to the opponent, suggesting a greater focus on the opponent's movements, while a positive value indicated the opposite. Experiment 1 was analyzed separately for the betrayal and unexpected help conditions (Fig 4B and 4D, bottom panels, S7A Fig), while Experiment 2 was analyzed by dividing the change in $F$ (the strength of betrayal or unexpected help) into seven equal intervals (Fig 6F). The gaze comparison value was averaged over the period from 1.5 to 3 seconds after the onset of the pursuit period (Fig 6G). To examine the relationship between observational focus during the pursuit phase and inference in the next trial, we calculated the Spearman correlation coefficient between gaze distance at the opponent (i.e., the strength of negative gaze comparison) and negative biases (perceived F - given F) (S7C Fig for experiment 1 and S7D Fig for experiment 2). We also calculated the correlation between how long participants gazed at the opponent and their negative bias (Fig 4G).

## Analysis of control stability

To assess participants' control stability over their avatar, we calculated jerk strength [63] (Fig 4E). Specifically, we computed the third derivative of the avatar's trajectory (i.e., jerk), squared it, and then averaged the result. Jerk strength for the prey and opponent's trajectories was measured similarly, along with the average distance between the gaze location and all three characters. Finally, we computed the partial Spearman's correlation coefficient between gaze distance to the self's avatar and jerk strength, controlling for gaze distance to the other characters. We also computed the correlation between the gaze distance to the participant's own avatar and the amount of reward earned on each trial (Fig 4F).

## Task-optimized RNN

To explore the relationship between gaze target shifts and inferential bias, we developed a task-optimized recurrent neural network (RNN) using the following steps (Fig 5A). In the first step, we trained a three-layer Long Short-Term Memory network (LSTM) to predict the F value from trajectory sequences. The model consisted of a sequence input layer (input size of 6, which was the x and y coordinates of three characters), an LSTM layer (64 hidden units), and a fully connected output layer (scalar output). It was trained on the inference phase of randomly selected 1000 trials, using the Adam optimizer (learning rate: 0.01, gradient decay: 0.9) with mean squared error (MSE) loss. The total MSE in the first step was computed as

$$MSE = \sum_{trial} Loss^2$$

where $Loss = F_{predict} - F_{actual}$

Gradients were iteratively computed and updated using accumulated values across trials to prevent the influence of any single trial from dominating the learning process. Participants' responses were not used for training the model.

In the second step, we examined how the neural network adapted to betrayal and unexpected help scenarios. To do this, the model trained in the first step was independently retrained 1,500 times as follows. The trajectories from the

pursuit phase were used as inputs instead of those from the inference phase, as it done in the first step. A total of 200 trials were used, with 100 randomly selected from betrayal trials and the remaining 100 from unexpected help trials, irrespective of participants. The selected set of 200 trials was changed for each re-training cycle within the 1500 training iterations. As a result, the $\tilde{F}$ value inferred by the RNN deviated from the given $F$ value in the corresponding inference phase. This deviation yielded MSE loss from a negative difference in the betrayal trials ($Loss^- = F_{predict} - F_{actual} < 0$) or a positive difference in the unexpected-help trials ($Loss^+ = F_{predict} - F_{actual} > 0$). The total loss from 200 trials was computed as follows;

$$MSE = \beta_{bet} \cdot Loss^- + (1 - \beta_{bet}) \cdot Loss^+$$

where $\beta_{bet}$ represents sensitivity to betrayal. When $\beta_{bet} = 1$, the model learns only from mean squared error (MSE) generated by negative differences trials (i.e., betrayal), while when $\beta_{bet} = 0$, it learns only from MSE generated by positive differences trials (i.e., unexpected help). $\beta_{bet}$ was initially set as neutral (=0.5). Next, with the recurrent and output layer weights fixed, the input layer weight and $\beta_{bet}$ were optimized using the same procedure as in Step 1, based on total MSE. This was to examine whether the RNN mimicked the observational focus shift observed in humans in the context of betrayal or unexpected help. For each of the re-learning iterations, the optimization process was repeated until $\beta_{bet}$ reached an extreme value (either above 0.9 or below 0.1). The changes in $\beta_{bet}$ across epochs are shown in Fig 5B, and the given F values from the inference phase used for training are shown in Fig 5C.

To assess potential overfitting in training, half of the trials were used as the training set (50 trials for the betrayal type and another 50 trials for the unexpected help type). The remaining trials, set aside separately, were used as the test dataset. The accuracy (1 - total loss) of both the training and test sets across epochs revealed that the test dataset's accuracy slightly declined from the 65th epoch, but the drop was negligible (~0.5%) (S8 Fig). Additionally, the tendency to categorize into two networks was already evident from the 65th epoch. By the 65th epoch, nearly all re-learning iterations had already reached the extreme value of $\beta_{bet}$. Therefore, the current results were not considered to be due to overfitting.

We examined the step-2 RNN in several ways. Firstly, the inferential bias of the RNN model was measured (Fig 5D). To achieve this, inference phase trajectories from step 1 training were provided as input again without additional weight optimization (i.e., forward pass test). The bias was calculated by subtracting the actual F value from the inferred $\tilde{F}$ value, following the same approach used in the choice analysis (shown in the subfigure of Fig 3A). We tested whether the changes in the activation of each unit were correlated with the inferential biases of the model. For interpretability, the sign of the weight change was reversed if it showed a negative Spearman correlation with the RNN's inferential bias. This was to ensure that the activation increase of an RNN unit consistently signaled a cooperative bias. We then compared the absolute weight changes associated with the opponent's position and the player's position (Fig 5E). To quantify the effect of each weight change on inference, we computed the vector norms of changes in the input weights for player- and opponent-related channels as the network transitioned from step 1 to step 2. We then related these weight changes to shifts in the inferred F value to assess the relative contribution of each input channel (Fig 5F).

In the third step, we examine the effect of observational focus shifts on sensitivity to betrayal (Fig 5G). To test this, we reversed the sign of the weights corresponding to the opponent's and player's avatar positions in the input layer. All other learnable parameters were kept fixed, and only $\beta_{bet}$ was re-optimized from its pre-reversal value using the same loss function as in the second step. Since the number of parameters to be learned has decreased, the learning rate was adjusted to 0.05. The changes in $\beta_{bet}$ across epochs are shown in Fig 5H, and the portion of $\beta_{bet}$ shifting from one extreme to the other (i.e., 0.1→0.9 or 0.9→0.1) is shown in Fig 5I.

## Statistical test

All pairwise comparisons between two values were conducted using two-tailed, independent t-tests. To examine interactions among three variables, we conducted a repeated measures ANOVA. We conducted a linear trend analysis to

evaluate how responses changed consistently across trials. For time series data, to account for the multiple comparison problems, we applied a nonparametric, cluster-based permutation approach to correct the p-value criterion for the two-tailed, one-sample t-test using MATLAB's statistical toolbox [64]. Specifically, this method grouped neighboring time points that showed similar effects into clusters, and a cluster-level t-value was calculated. We then shuffled the condition labels many times and recomputed these statistics to create a permutation distribution. By comparing the observed cluster to this distribution, we determined whether the effect was significant while properly controlling for multiple comparisons.

## Supporting information

**S1 Fig. Extended results of Experiment 1.** (A) The psychometric curves of F values for the forty participants. The solid line represents the cumulative Gaussian fit, while the dashed lines indicate the data. (B) Replication of the main effect in eight participants recruited in a separate session, showing the same pattern of results. (C) Biases in the inference are grouped by sex. (D) No correlation was found between participants' age and biases in the inference. (E) Negative biases in perceived F value increased with repeated betrayal but did not after unexpected help. Stars indicate statistical significance (***, $p < 0.001$; n.s., not significant). (F) Biases in perceived F value re-aligned at the moment of block change. Colors indicate the block type before the block change occurred. Error bars and shaded ribbons represent ±1 SEM, and dots indicate individual participants.
(TIFF)

**S2 Fig. Reward magnitude had little effect on perceived F.** (A) Perceived F (i.e., the probability of choosing the boost item on the subsequent trial) as a function of normalized reward magnitude on the previous trial. The slope reflects each participant's sensitivity to the reward from the previous trial. (B) Relationship between the slopes for low-reward and high-reward trials across participants. (C) Mean perceived F following betrayal and unexpected-help conditions within the just-noticeable-difference range. Error bars and shaded regions represent ±1 SEM; dots indicate individual participants.
(TIFF)

**S3 Fig. Basic features comparison between betrayal and unexpected-help conditions.** The position, speed, and acceleration of the player, prey, and opponent, as well as pairwise distances and movement angles among the three agents, were compared between trials following betrayal and trials following unexpected help. Crosses indicate mean ±1 SEM. 'fr' indicates frames (60 Hz sampling).
(TIFF)

**S4 Fig. Decision time did not affect inferential bias.** (A) No difference in decision times during the item decision phase between betrayal and unexpected help conditions. (B) Bias in perceived F as a function of decision time. Error bars and shaded ribbons represent ±1 SEM, and dots indicate individual participants.
(TIFF)

**S5 Fig. Simulation of parameters in the auto-regressive logistic regression model.** The weight on the experience of betrayal or unexpected help ($\beta_{hys}$) and the weight on the three avatars' trajectories ($\beta t$) were parametrically amplified or attenuated. Psychometric curves based on logistic regression model data capture inference during the last second before the item decision phase. Changes in $\beta_{hys}$ influenced the overall baseline difference between the betrayal and unexpected help conditions, while changes in βt affected the inferential bias in a given context. Shaded error bars represent ±1 SEM.
(TIFF)

**S6 Fig. Certainty of inference did not differ across conditions in the energy landscape.** (A) The depth (difference between minimum and maximum values) and steepness (slope) of the inference energy landscape were similar across

the basins of betrayal and unexpected help. (B) d′ values did not differ across conditions. (C) The decision criterion was significantly more conservative after betrayal. Error bars represent ±1 SEM, and dots represent individual participants. (TIFF)

**S7 Fig. Rejection of possible confounds in observation results of Experiment 1.** (A) Comparison of player and opponent gaze during the pursuit period, restricted to trials with matched F values (0.4 < F < 0.6). (B) Eight participants recruited in a separate session showed the same gaze-shift pattern. (C-D) Spearman's correlation between gaze directed at the opponent and competitive bias in perceived F on the subsequent trial in Experiments 1 (B) and 2 (C). Error bars and shaded ribbons indicate ±1 SEM, and dots represent individual participants.
(TIFF)

**S8 Fig. Accuracy of F value prediction by the task-optimized RNN during step 2.** Black: training dataset (half of the dataset). Yellow: the test dataset (the other half). Shaded error bars represent ±1 SEM.
(TIFF)

**S9 Fig. Reconstructed energy landscapes of inference.** Separate energy landscapes are shown for trials with low (F < 0.33) and high F value (F > 0.67) (A), along with their corresponding stable fixed points (lowest points of the basins) in the phase diagram (B). Error bars represent ±1 SEM. Solid lines indicate the increasing betrayal phase, while dashed lines represent the decreasing phase.
(TIFF)

**S1 Video. Example of betrayal scenario in Experiment 1. After the inference phase, the participant selected the boost item but subsequently lost the prey to the opponent during the pursuit phase. For illustrative purposes, the opponent's F value was altered from 1 during the inference phase to 0 in the pursuit phase to depict the most extreme behavioral shift. In the actual experiment, however, the maximum change in F (ΔF) was -0.45.**
(MP4)

**S2 Video. Example of unexpected help scenario in Experiment 1. During the pursuit phase, the participant captured the prey with ease as the opponent drove it toward the player. For illustrative purposes, the opponent's F value was altered from 0 during the inference phase to 1 in the pursuit phase to depict the most extreme behavioral shift. In the actual experiment, however, the maximum change in F (ΔF) was + 0.45.**
(MP4)

**S1 Table. Logistic regression coefficients during Experiment 1. t1-t5 represent successive 200-ms time bins of the three-character trajectories.**
(DOCX)

**S2 Table. Logistic regression coefficients during Experiment 2. t1-t5 represent successive 200-ms time bins of the three-character trajectories.** Inc and Dec refer to the betrayal-increasing and betrayal-decreasing phases, respectively.
(DOCX)

## Author contributions

**Conceptualization:** Sangkyu Son, Seng Yoo.

**Data curation:** Sangkyu Son.

**Formal analysis:** Sangkyu Son.

**Funding acquisition:** Seng Yoo.

**Writing – original draft:** Sangkyu Son, Seng Yoo.

**Writing – review & editing:** Sangkyu Son, Seng Yoo.

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
