## [Decision Letter · Decision Letter 0]

30 Sep 2025

PCOMPBIOL-D-25-01175

Selective Observation Under Limited Resources Biases Social Inference Through Hysteresis

PLOS Computational Biology

Dear Dr. Yoo,

Thank you for submitting your manuscript to PLOS Computational Biology. After careful consideration, we feel that it has merit but does not fully meet PLOS Computational Biology's publication criteria as it currently stands. Therefore, we invite you to submit a revised version of the manuscript that addresses the points raised during the review process.

While the reviewers found the topic of your manuscript engaging, several concerns were raised regarding methodological details. In particular, (i) the data have not been deposited in a repository as claimed, and (ii) the study was not preregistered, raising questions about replicability. After discussion with the section editor, we ask that you deposit your data in a publicly accessible repository (not necessarily GitHub) and, although preregistration is not mandatory for PLOS Computational Biology, include additional data or analyses to demonstrate the robustness and replicability of your findings. Please also address all other reviewer comments in your revision.

Please submit your revised manuscript within 60 days Nov 30 2025 11:59PM. If you will need more time than this to complete your revisions, please reply to this message or contact the journal office at ploscompbiol@plos.org. Please include the following items when submitting your revised manuscript:

We look forward to receiving your revised manuscript.

Kind regards,

Lusha Zhu, Ph.D.

Academic Editor

PLOS Computational Biology

Hugues Berry

Section Editor

PLOS Computational Biology

**Journal Requirements:**

3) We notice that your supplementary Figures are included in the manuscript file. Please remove them and upload them with the file type 'Supporting Information'. Please ensure that each Supporting Information file has a legend listed in the manuscript after the references list.

Potential Copyright Issues:

i) Figure 3C. Please confirm whether you drew the images / clip-art within the figure panels by hand. If you did not draw the images, please provide (a) a link to the source of the images or icons and their license / terms of use; or (b) written permission from the copyright holder to publish the images or icons under our CC BY 4.0 license. Alternatively, you may replace the images with open source alternatives. See these open source resources you may use to replace images / clip-art:

2) If any authors received a salary from any of your funders, please state which authors and which funders..

**Reviewers' comments:**

Reviewer's Responses to Questions

**Comments to the Authors:**

Reviewer #1: This study investigated how the choice of observational target biases inference about others’ intentions. The authors showed that participants were able to modulate their behavior depending on their inference about the opponent’s strategy (cooperative or competitive). Furthermore, this inference was biased when the opponent had acted more competitively than expected in the previous interaction. Using eye-tracking, the authors also found that the bias became stronger when participants chose to observe the opponent instead of their own avatar and this choice came at the cost of reduced control over their own avatar. Finally, they reproduced the behavioral patterns using a recurrent neural network (RNN) model.

I am wondering whether the study adequately addresses the main research question: How does choosing what to observe bias inference about others’ intentions? (e.g., lines 17–18). The authors clearly demonstrated that participants’ decision-making about whether to boost or hinder the opponent’s speed (P(boost)) was modulated by the opponent’s intention (F) to cooperate or compete. However, this promising finding is not directly related to the allocation of limited observational resources (i.e., selective observation).

The eye-tracking data may be somewhat relevant to the main research question. However, the corresponding description (lines 240–252) is overly speculative. That is, the conclusions drawn are not directly supported by the data and rely heavily on speculation. I do not question the quality of the data, but it remains unclear how the observed gaze patterns are related to the biased inference about others’ intentions and social decision-making.

The authors used P(boost) as a proxy for the perceived intention of the opponent, inferred from observing their actions. Based on the result that P(boost) was modulated by F, they conjecture that participants inferred the opponent’s intention. However, I wonder whether and how they can rule out alternative explanations. For example, participants may have inferred the opponent’s ability to catch the prey, specific actions, or other attributes rather than intention per se.

Likewise, although the authors claim that an opponent’s behavior in the previous interaction biases social inference about their intention, an alternative interpretation is that participants’ reaction to the inferred intention—rather than the inference itself—was biased. Is it possible to rule out this interpretation?

Ideally, preregistration would have been conducted. This is not to suggest that exploratory analyses are inherently problematic. Moreover, it is unclear whether the authors are conducting exploratory or confirmatory analyses. If the analyses are confirmatory, preregistration would have been important; if exploratory, it would be valuable for the authors to clearly state this.

It is also unclear how the RNN analysis contributes to understanding the underlying mechanism. While it is not surprising that a complex model like an RNN can reproduce behavioral data, it remains unclear what kind of understanding or interpretation this provides. Does it help constrain specific interpretations? For example, the statement “These RNN results demonstrate that reallocating observational resources to the opponent’s movement mainly drives the inferential bias toward negative F values, revealing the source of the asymmetrical bias observed in human participants” is plausible. However, while this may be a relevant interpretation, can alternative interpretations be excluded? Given the complexity of RNN, there might be so many possible architectures that can reproduce the data. Why and how did the authors choose the specific one?

Moreover, I do not agree with the authors’ claim that RNN modeling can test causality (lines 100–102 and 292). The authors should carefully distinguish between causality in RNN simulations and causality in human decision-making.

In lines 157–161, the authors compared the magnitudes of monetary gain and loss, and concluded that betrayal did not cause greater monetary loss than the gain from unexpected help. However, psychological impacts of monetary loss can outweigh gains, as suggested by Prospect Theory in behavioral economics.

How were the sample sizes in Experiments 1 and 2 determined?

Regarding reward payment, why was the original score multiplied by the opponent’s F value?

It would be helpful if the authors provided detailed information about the cluster-based permutation approach used to correct the p-value criterion.

Is Fig. 1 directly relevant to the specific context of this study?

Reviewer #2: This manuscript developed an interactive prey-pursuit task where participants needed to infer an opponent’s hidden intentions through observation to make decisions. They found that participants’ behavior was history-dependent, where they overestimated the opponent's competitiveness after the opponent acted more competitively than expected, but no such bias when the opponent was more cooperative than expected. Further, the effect of betrayal history was greater when betrayal was an increasing phase than when it was a decreasing phase. To understand the bias, they used eye-tracking and found different observing patterns when participants experienced betrayal and unexpected help. Then they trained a task-optimized RNN and showed that shifting the focus of observation can reverse inference biases.

This paper is an insightful contribution to the field. The authors proposed a novel task which allowed inferring others’ intentions through a dynamical, continuous process and real-time interaction, which has potential to explore a broad range of social problems such as coordination and game. And they introduced the energy landscape framework, which provided new insight for understanding decision making process. Their results are comprehensive. However, we have identified several issues that need to be addressed to strengthen the results and clarify the conclusions. Further details on our concerns can be found below.

1. The authors defined perceived/inferred friendliness as the estimated choice probability by autoregressive logistic regression, and most of the results were based on this index. But the regression details are missed, including the precise regression equation, detailed regression results (table with regressors and the corresponding statistics), and estimation/optimization details. They are important to support the model. Besides, whether the results remain unchanged when using real boost probability instead of the regression estimation? At least some model-free results should be shown to validate the conclusions and analyses.

2. Did the patterns of response time during decision making have any differences after experiencing betrayal or unexpected help under different F levels? And the relationship between response time and perceived F is also needed to be checked.

3. Lack of formal specification (mathematics) of the energy calculation and RNN loss function.

Reviewer #3: Summary

In this study, the authors introduce a novel two-agent hunting paradigm to investigate how inferences about others' motives are shaped by experience and selective observation. They found that interacting with opponents that behaved more competitively than expected selectively and negatively biased subsequent inferences regarding their cooperativeness. Eye gaze analyses suggest that subjects allocated more attention to the other's actions during such betrayals, which in turn was linked to increased bias in subsequent trials. They further characterized this bias with an energy landscape approach, and used an RNN to claim a causal effect of attention on the bias. In a second experiment, they provide evidence that this effect propagates across trials in a hysteresis-like fashion.

The manuscript addresses a relevant question using an innovative task and an impressive array of analyses. The text is well-written, logically structured, and the large number of high-quality figures transparently portray the data and nicely illustrate all main analyses, which is really helpful. The findings are interesting and the conclusions are largely well supported by the data. However, three major points (and a few minor ones) should be addressed before publication: The results raise one main question that currently remains unaddressed; the RNN procedures seem to (at least in parts) predispose their outcomes; and some of the findings in Exp. 1 seem to be at odds with the conclusions from Exp. 2.

Major comments

#1) Given the center-stage and novelty of the asymmetric inference bias, a central question that surprisingly remains unaddressed in results and discussion is where this bias originates. While the authors rule out larger monetary consequences as a driver (via the analysis in Fig 2G), could loss aversion change this conclusion? Relatedly, could betrayal aversion or similar psychological motives be at play? On the other hand, the fact that the RNNs seem to recreate the asymmetry speaks to reasons other than psychological ones. Can the authors provide any such reasons, or at least speculate about them?

#2) If I understand the RNN analyses correctly, the way the training is set up in step 2 explicitly trains the models to make asymmetrically biased judgments. Specifically, if the loss used for training is the difference between the F value as predicted by the RNN (for the input sequence), and the value from the corresponding inference phase (which does not seem to enter the model again and hence is not available to the network), the *only* way to improve the prediction is by introducing a bias. As this can’t work for an equal number of positive and negative prediction error trials, the network is forced during training to bifurcate to only focus on one type of errors (via adjusting the beta parameter). Is this understanding correct? If so, the authors should revise the corresponding sections to better reflect this. However, even if it is the case (or possibly even more so), it’s still interesting and unclear why the bias only seems to emerge in the betrayal nets (which relates to #1 above).

#3) The findings from Exp. 2 are summarized to suggest that “stopping the betrayal will not immediately restore neutral inference due to its momentum [when the degree of betrayal continues to increase].” However, in Exp. 1, the authors report: “[...] repeated betrayal progressively reinforced the bias toward more competitive inferences (S1D Fig; linear trend analysis [...]). However, the biases after the betrayal experience immediately returned to a neutral state when the betrayal ceased (i.e., at block change) (S1E Fig).” How can these two findings be reconciled - shouldn’t momentum in Exp. 1 prevent an immediate return to a neutral state (even if it’s not increasing as in Exp. 2)?

Minor comments

Choice of title/abstract: The authors report two main findings: Exp. 1, and the majority of analyses and figures, are concerned with characterizing the asymmetry of the history-dependent inference bias which results from the very last trial, while Exp. 2 is concerned with effects that propagate across trials in a hysteresis-like fashion. This makes one wonder if the title should really focus on the latter, as it currently does, rather than the former. Relatedly, the authors might consider mentioning the energy landscape and/or RNN analyses in the abstract, in order to attract the right target audience.

Figure 2E: Is it possible that the top two panels are flipped? The top-left one looks as if the opponent is going straight for the prey (i.e., being competitive), whereas the top-right one looks more like herding.

Figure 2F: "hindering a low-F opponent was more beneficial" - but hindering seems to generally just render the opponent’s F inconsequential, as the reward does not seem to be affected by F (whereas the effect of boosting changes as a function of F); is this because the other's behavior simply becomes irrelevant when hindered?

Figure 3A: Please state that the figure is across participants, and refer to the corresponding Supplementary Figure for individual subjects.

Line 232 ff: Please also report effect sizes in the main text (which are depicted in Figure 4C/D); especially given that they are rather small.

Line 307: “As in Experiment 1, stronger betrayal experiences in previous trials led to a greater bias toward” - is this dose-response relationship really shown? I thought Exp. 1 always compared betrayal vs. unexpected help at t-1, whereas Exp. 2 compared increasing vs. decreasing betrayal (which, if anything, rather suggests that stronger betrayal in previous trials, as experienced in the decreasing phase, lead to *smaller* biases).

Line 281: "These RNN results demonstrate that reallocating observational resources to the opponent's movement mainly drives the inferential bias”, but before that it is stated that “a decrease of the input weight led to a corresponding inference bias”. How can a *de*crease in weights be interpreted as allocating *more* resources?

Relatedly, can the authors speculate on why more/less focus on the other (by humans and RNNs) only affects inferences regarding competitiveness? This relates to the major comment #1 above, but with emphasis on the causal effect on attention.

Typos/wording:

281: “F value” seems to be accidentally added.

626: “by following the steps [?] (Fig 5A)” end of the sentence seems to be missing.

652: “unexpected [help?] trials”

718: “During the inference phase [...]” -> “Across trials, the F value during the inference phase [...]” would be correct, as there is no change *during* the inference phase, right?

731: “resembles” -> “can be represented as” would be more appropriate.

733: “moving-windowed trajectories [during the inference phase]” would be more explicit.

Reviewer #4: Son and Yoo have developed an interesting new behavioural task, in which hidden intentions of a computer opponent can be deduced (i) by past interactions and (ii) by selectively allocating one’s eye gaze to the spatial trajectory of this opponent. Both participants’ choices and their eye gaze were related to the fluctuating “friendliness” (i.e., cooperativeness) of the agent. The authors used an autoregressive logistic regression model and developed a RNNs to describe and mimic participants’ behavior.

Specifically, participants (depicted as a green dot on a screen) could chase prey (depicted as a blue square). Crucially, an opponent (depicted as red dot) could help or hinder. The intention of the opponent varied from cooperative to competitive (manipulated via the parameter F for friendliness). Depending on the opponent’s intention, participants should “boost” or “hinder” the opponent’s speed in order to maximize their own probability to catch the prey. Each trial had two phases: Inference and pursuit. Unbeknownst to the participants, the intention of the opponent changed between these two phases resulting in two conditions: betrayal (i.e., cooperative to competitive) and unexpected help (i.e., competitive to cooperative). These two conditions changed probabilistically in a block-wise fashion in Experiment 1 (n=54) and in a “decreasing/increasing fashion” in Experiment 2 (n=14). Results showed overall lower boost decision after betrayal trials than after unexpected help trials. The inferred F (in regression models) differed from the true F in the betrayal trials but not the unexpected help conditions. Gaze on the opponent also differed between these conditions.

In my view, the paper is highly innovative and overall well-written. Please see below for my comments and questions.

1. Trial & block structure & changes in F: This was not entirely clear to me and is also one of the reasons why I have tried to summarize the experimental setup in such detail above. A few related questions:

a. What was the motivation for changing F between inference and pursuit phases? Why was there an inference phase at all? That is, why exactly did the authors need a phase in which participants could not act? Could the change in F not be implemented ONLY between consecutive trials?

b. Crucially, why was there a block-wise change in the sign of the change of F? For many of the analyses it is unclear whether the authors looked at trial-by-trial differences or block-by-block differences. I think they mostly looked at the former but then why is it necessary to have block-wise changes?

c. F changes quite quickly (Fig 2D upper part). Is this truly necessary? This makes learning very hard. What would the authors and what would their models predict for slower changes? It could be that the bias would disappear.

d. Crucially, the definition of betrayal vs. unexpected help conditions seems to only depend on the sign of the change in F between inference and pursuit phases but not the “ground level” of F in the inference phase. I would like to see some of the analyses that the authors did with deltaF with F_inference and F_pursuit as separate predictors.

e. The authors should show the distribution of these F values. For example, Fig 3A shows binned data but it is unclear how many data points are within each bin. I guess the middle ranges of F contain more data points.

2. The “energy landscape analyses” were also not quite clear to me:

a. The methods section has more details but the following description in the results is not quite enough for the results section. “Thus, the model incorporated two key parameters: βt, which captures how well participants infer the opponent’s intentions from the trajectories of the three characters during the inference phase, and βhys, which regulates how past experiences of betrayal or unexpected help influence inference bias. In sum, βt primarily alters the steepness of the psychometric curve, while βhys primarily modulates the magnitude of inference bias (S2 Fig).“ I found the following part in the methods helpful “To construct this, we computed the pairwise Euclidean distances among the three characters at each time point over a 1s period, then averaged them within 200-ms bins, resulting in 15 representative feature values (3 characters × 5 time points).

b. Also, Fig 3D does not quite make it clear. This is easy to improve.

c. Why did the authors not look into the three different trajectories for these analyses? Relatedly, we did they not look at the trajectory of the prey for the gaze data (see comment below)?

d. Why are only three time points plotted in Fig 3E and not the 5 time points analysed?

e. I did not understand the following: “To analyze the meaning of each regressor on intention inference, we conducted simulations by multiplying 0, 1/3, 2/3, or 1 on βt, or multiplying 0, 1, 2, or 3 on βhys.”

3. Eye-tracking:

a. My main point here is what are the distances to the prey? Are some of these results spurious in the sense that participants focus more on the prey which is influenced by the F value of the opponent?

b. The correlations very small. Could the authors comment on this and try to situate the explained variance?

c. I am not quite convinced by the costs in terms of “jerk strength.“ Did this result in fewer catches of the prey?

d. The following section in the results seems to me too speculative. It might be ok in the discussion. “Overall, these trade-offs suggest that participants prioritized what to observe based on cost; the opponent's betrayal was more costly than losing some control over their own avatar. This cost driven observation became a natural source of information asymmetry. That is, by shifting their gaze tothe opponent, participants gained more information about unexpected competitiveness; otherwise, they learned nothing new. Our results suggest that this difference in information from selective observation eventually biased the later inferences.”

4. RNN: I am not an expert on RNNs and therefore I cannot comment on the details. But I definitely do not like the word “causal” in the following sentence: “To verify whether gaze target shifts causally influence inferential bias, we developed a task optimized recurrent neural network (RNN)”

5. Fig 1 on POMDP. I know more about POMDPs than on RNNs. I generally like the background given in Fig 1 but the authors did not use a POMDP to simulate participants’ data.

a. In Fig 1, it is slightly confusing that capital “I” is used both for others’ Intentions and self’s Inference.

b. It is also slightly confusing that the observations of the other are included. Wouldn’t that be seeing the others’ eye gaze in this setup? But here the opponent clearly has no eye gaze.

c. I like that colour code seems to correspond to the task layout. But then “green” should be for the self’s actions.

6. Task design & additional task changes due to F:

a. Did participants know this and did they see this on a trial-by-trial basis? “To prevent participant demotivation, if the opponent caught the prey before the player, the participant received a small portion of the score they would have earned by catching it themselves. Specifically, the original score was multiplied by the opponent's F value and 0.5.“

b. Was the following really necessary? “To encourage accurate predictions of the opponent's F value during the item decision period, the player's speed in the pursuit period was slightly adjusted based on decision accuracy. Specifically, when the opponent's behavior was predictable (F = 1 or F = 0) and the choice was correct, the player's speed increased to match the prey's maximum speed. Conversely, if the choice was incorrect, the speed difference from the prey could increase up to twice the original gap. When the opponent's behavior was ambiguous (F = 0.5), the player's speed remained unchanged regardless of the participant's choice.

7. A potential strategy: As far as I understand, the authors assume that the participants maximize their gains by trying to catch the prey during the pursuit phase. But participants might also try to explicitly check what the opponent does when they move away from the prey. Could such behavior be seen?

8. “The magnitude of F change was randomly set between +/-0.35 and 0.45.“ Should this be “between -0.45 and 0.45”?

9. Typo in line 739: “F > 0.76“ should be “F > 0.67“

I did not check the code at this stage.

**Have the authors made all data and (if applicable) computational code underlying the findings in their manuscript fully available?**

The PLOS Data policy requires authors to make all data and code underlying the findings described in their manuscript fully available without restriction, with rare exception (please refer to the Data Availability Statement in the manuscript PDF file). The data and code should be provided as part of the manuscript or its supporting information, or deposited to a public repository. For example, in addition to summary statistics, the data points behind means, medians and variance measures should be available. If there are restrictions on publicly sharing data or code —e.g. participant privacy or use of data from a third party—those must be specified.requires authors to make all data and code underlying the findings described in their manuscript fully available without restriction, with rare exception (please refer to the Data Availability Statement in the manuscript PDF file). The data and code should be provided as part of the manuscript or its supporting information, or deposited to a public repository. For example, in addition to summary statistics, the data points behind means, medians and variance measures should be available. If there are restrictions on publicly sharing data or code —e.g. participant privacy or use of data from a third party—those must be specified.

Reviewer #1: **No:**I could not find the data in https://github.com/SangkyuSon/socialObservationHysteresisI could not find the data in https://github.com/SangkyuSon/socialObservationHysteresis

Reviewer #2: None

Reviewer #3: **No:**Code is available, but data does not seem to be.Code is available, but data does not seem to be.

Reviewer #4: Yes

PLOS authors have the option to publish the peer review history of their article (what does this mean?). If published, this will include your full peer review and any attached files.). If published, this will include your full peer review and any attached files.

.

Reviewer #1: No

Reviewer #2: No

Reviewer #3: No

Reviewer #4: No

**Figure resubmission:**
---

## [Decision Letter · Decision Letter 1]

19 Feb 2026

PCOMPBIOL-D-25-01175R1

Selective Observation following Betrayal shapes the Social Inference Landscape

PLOS Computational Biology

Dear Dr. Yoo,

Thank you for submitting your manuscript to PLOS Computational Biology. After careful consideration, we feel that it has merit but does not fully meet PLOS Computational Biology's publication criteria as it currently stands. Therefore, we invite you to submit a revised version of the manuscript that addresses the points raised during the review process.

The reviewers agree that the manuscript has been substantially improved and recommend minor revisions prior to acceptance. Please clarify in the manuscript the rationale for the addition of participants and any resulting changes to sample size, analyses, or reported statistics, and clearly distinguish between samples where applicable. In addition, please incorporate into the Methods or Supplementary Information the methodological explanations provided in the response letter and clarify if/how changes of analyses affected the results and conclusions of the paper.

We look forward to receiving your revised manuscript.

Kind regards,

Lusha Zhu, Ph.D.

Academic Editor

PLOS Computational Biology

Hugues Berry

Section Editor

PLOS Computational Biology

**Journal Requirements:**

1) Please ensure that the Title in your manuscript file and the Title provided in your online submission form are the same.

2) We have noticed that you have uploaded Supporting Information files, but you have not included a complete list of legends. Please add a full list of legends for your Supporting Information files after the references list.

3) Please amend your detailed Financial Disclosure statement. This is published with the article. It must therefore be completed in full sentences and contain the exact wording you wish to be published.

2) If any authors received a salary from any of your funders, please state which authors and which funders..

4) We notice that your supplementary Tables are included in the manuscript file. Please remove them and upload them with the file type 'Supporting Information'. Please ensure that each Supporting Information file has a legend listed in the manuscript after the references list.

5) Please make sure to upload supplementary figures as supplementary files, instead of categorizing them as figure types in the file inventory.

**Reviewers' comments:**

Reviewer's Responses to Questions

**Comments to the Authors:**

Reviewer #1: The authors have addressed all of my concerns.

Reviewer #2: Thank you for submitting the revised version of the manuscript. I have reviewed the authors' responses to my previous comments and the corresponding modifications made to the paper. I am pleased to confirm that all the concerns and questions I raised have been adequately addressed and resolved. The revisions have significantly enhanced the clarity, accuracy, and overall quality of the work. Based on this, I recommend acceptance of the manuscript in its current form.

Reviewer #3: The authors have substantially improved the manuscript, in particular strengthened the presented evidence with further analyses and improved the clarity and transparency of their methods. However, the following two points should still be addressed before publication.

1) The novel control analyses for psychological factors are helpful to rule out several alternative explanations. However, as it stands, the analysis regarding loss aversion is not quite clear to me and should be further elaborated: How exactly does this capture loss aversion? Isn't reward known only _after_ a trial and hence cannot be a modulating factor (i.e., I cannot fear what I don't know)? Also, Fig 2F shows that reward is a monotonic function of given F, so shouldn't perceived F also be (at least weakly) positively linked to reward?

2) It appears that the sample size from study 1 and various test statistics have been altered, but not highlighted in the manuscript. Can the authors provide a rationale why these additional subjects were added, and state if any findings and conclusions from the paper were affected by including them? One response to reviewer mentions replications, but the subjects seem to have been added to study 1 rather than as separate replication sample. Relatedly, the different samples should be clearly separated throughout the manuscript (e.g., the Results section starts by reporting the total N, but then goes on to report results based only on the subjects from study 1).

Reviewer #4: The authors have answered all my questions. My suggestion for minor revisions is that they should also include the explanaitons for the design choices and the methodological aspects into

**Have the authors made all data and (if applicable) computational code underlying the findings in their manuscript fully available?**

The PLOS Data policy requires authors to make all data and code underlying the findings described in their manuscript fully available without restriction, with rare exception (please refer to the Data Availability Statement in the manuscript PDF file). The data and code should be provided as part of the manuscript or its supporting information, or deposited to a public repository. For example, in addition to summary statistics, the data points behind means, medians and variance measures should be available. If there are restrictions on publicly sharing data or code —e.g. participant privacy or use of data from a third party—those must be specified. requires authors to make all data and code underlying the findings described in their manuscript fully available without restriction, with rare exception (please refer to the Data Availability Statement in the manuscript PDF file). The data and code should be provided as part of the manuscript or its supporting information, or deposited to a public repository. For example, in addition to summary statistics, the data points behind means, medians and variance measures should be available. If there are restrictions on publicly sharing data or code —e.g. participant privacy or use of data from a third party—those must be specified.

Reviewer #1: Yes

Reviewer #2: None

Reviewer #3: **No:**Data still does not seem to be availableData still does not seem to be available

Reviewer #4: None

PLOS authors have the option to publish the peer review history of their article (what does this mean?). If published, this will include your full peer review and any attached files.). If published, this will include your full peer review and any attached files.

**Do you want your identity to be public for this peer review?** For information about this choice, including consent withdrawal, please see our  For information about this choice, including consent withdrawal, please see our Privacy Policy..

Reviewer #1: No

Reviewer #2: No

Reviewer #3: No

Reviewer #4: **Yes:**Christoph W. KornChristoph W. Korn

**Figure resubmission:**
---

## [Editor Report · Decision Letter 2]

6 Apr 2026

Dear Dr Yoo,

We are pleased to inform you that your manuscript 'Selective Observation following Betrayal shapes the Social Inference Landscape' has been provisionally accepted for publication in PLOS Computational Biology.

Best regards,

Lusha Zhu, Ph.D.

Academic Editor

PLOS Computational Biology

Hugues Berry

Section Editor

PLOS Computational Biology

---

## [Editor Report · Acceptance letter]

PCOMPBIOL-D-25-01175R2

Selective Observation following Betrayal shapes the Social Inference Landscape

Dear Dr Yoo,

I am pleased to inform you that your manuscript has been formally accepted for publication in PLOS Computational Biology. Your manuscript is now with our production department and you will be notified of the publication date in due course.

With kind regards,

Aiswarya Satheesan
